

# On computing non-equilibrium dynamics following a quench

Neil J. Robinson⋆, Albertus J. J. M. de Klerk† and Jean-Sébastien Caux‡

Institute for Theoretical Physics, University of Amsterdam,
Postbus 94485, 1090 GL Amsterdam, The Netherlands

⋆ neil.joe.robinson@gmail.com , † a.j.j.m.deklerk@uva.nl , ‡ j.s.caux@uva.nl

## Abstract

Computing the non-equilibrium dynamics that follows a quantum quench is difficult, even in exactly solvable models. Results are often predicated on the ability to compute overlaps between the initial state and eigenstates of the Hamiltonian that governs time evolution. Except for a handful of known cases, it is generically not possible to find these overlaps analytically. Here we develop a numerical approach to preferentially generate the states with high overlaps for a quantum quench starting from the ground state or an excited state of an initial Hamiltonian. We use these preferentially generated states, in combination with a "high overlap states truncation scheme" and a modification of the numerical renormalization group, to compute non-equilibrium dynamics following a quench in the Lieb-Liniger model. The method is non-perturbative, works for reasonable numbers of particles, and applies to both continuum and lattice systems. It can also be easily extended to more complicated scenarios, including those with integrability breaking.

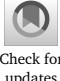

# 1 Introduction

Non-equilibrium strongly correlated systems have been the subject of intense study over the last decade [1–10]. Spurred on by experiments in ultra-cold atomic gases [11–13], questions of a fundamental nature have taken the center stage: What general principles govern properties of a non-equilibrium system? Are there non-equilibrium phases that have no equilibrium analogue? How does an isolated quantum system equilibrate and thermalize when time evolution is unitary? In the process of addressing such questions, it was realized that conservation laws play a central role in the description of non-equilibrium physics, strongly restricting the dynamics that can occur and the processes that govern equilibration [6, 14–21]. Nowhere is this more evident than in integrable models, where the presence of an extensive number of local conservation laws leads to an absence of thermalization [14, 15].

Integrable quantum many-body systems may, at first glance, appear to be little more than an academic curiosity. One may imagine that for an extensive number of local conservation laws to exist, there must be an extreme fine-tuning of a many-body Hamiltonian, and hence there is little chance of them being realized in an experimental system. Fortunately, this is not

the case. Perhaps the simplest non-trivial example is the Lieb-Liniger model [22–24] of delta-function interacting bosons confined to a single spatial dimension, which is almost perfectly realized in many cold atomic gas experiments (see, e.g., Ref. [11,25–27]), including those that probe non-equilibrium dynamics. Thus integrability, and the influence of conservation laws, can be directly examined in experiment.[1]

As a result, non-equilibrium dynamics of the Lieb-Liniger model has received a significant amount of theoretical attention [34–58]. In many of these studies the system is driven out of equilibrium via a quantum quench [59] of the interaction strength. Analytical studies have focused on cases where the initial states are eigenstates of the Lieb-Liniger model with either $c = 0$ or $c = \infty$, due to simplifications (both cases being 'non-interacting' in nature) that allow one to explicitly compute overlaps between the initial state and eigenstates of the Hamiltonian governing time evolution. With these overlaps at hand, expectation values of local operators in the long-time limit can be computed via, for example, the quench action method [5,19]. Generally, accessing the real-time dynamics of observables is still an outstanding challenge – away from the mentioned special initial states it is not known how to proceed. Analytically, one does not know how to compute the overlaps, a crucial ingredient for existing approaches, whilst numerically it is tough to deal with continuum models in a rigorous and well-controlled manner. Brute force computations, using the coordinate Bethe ansatz, are limited to very small numbers of particles, $N \sim 5$, and scale super-exponentially as $\propto (N!)^2$ without additional approximations [41,45].

In this work, we develop a novel numerical approach, motivated by the truncated spectrum approach [60], that allows one to compute overlaps. Our algorithm allows us to efficiently express the initial state in terms of the most important eigenstates of the Hamiltonian governing time evolution. With these overlaps at hand, we can then study the structure of the overlaps away from analytically tractable limits, compute real-time dynamics, and access the long time limit via the diagonal ensemble [15]. Here we present proof-of-principle computational results for interaction quenches in the Lieb-Liniger model for a reasonably small number of particles (although, we note, well beyond the reaches of 'brute force' coordinate Bethe ansatz computations [41,45]).

The truncated spectrum approach has recently been used to compute overlaps and non-equilibrium dynamics in both the Ising field theory [61–65] and the sine-Gordon model [66–69]. There are some fundamental differences between the approach taken here and those previously considered. We will explicitly work with a *strongly correlated computational basis* formed from eigenstates of an *interacting* integrable quantum system. Strong correlations are then inherently built into the basis, in contrast to the non-interacting bases of the Ising and sine-Gordon models. Furthermore, our approach abandons the conventional, energy-ordered, Hilbert space truncation metric and instead we are able to *preferentially target* the states contributing the largest overlaps. Our approach reduces the computational cost of calculations by orders of magnitude, as we will see below.

## 1.1 Layout

In Sec. 2 we introduce the system that we study, the Lieb-Liniger model, and its exact Bethe ansatz solution. We also discuss our quench protocol and formulation of the quench problem in terms of a perturbed Hamiltonian. Following this, in Sec. 3 we first describe the "ideal" numerical solution to the quench problem, and then discuss the development of the high overlap states truncation scheme (HOSTS) – an attempt to construct precisely this. To do so, we

---

[1]This is despite the fact that integrability is broken, albeit weakly, in experiments. The timescales for observing integrability breaking can be anomalously long, with the system exhibiting so-called 'prethermalization' [8,28–31], where the proximity to an integrable point still strongly restricts the dynamics. At very long times the system is still expected to thermalize [31–33].

describe how the basic truncated spectrum approach works and apply it to the problem. This reveals that: (i) the traditional truncated spectrum approach is not well-suited to the problem; (ii) numerical renormalization group extensions of this method are also not well-suited to the problem. The numerical renormalization group results give us inspiration for an alternative algorithm, based upon a better "ordering metric" for the Hilbert space truncation. We explore this and, putting all these results together, can construct the initial state to reasonable accuracy at some (not insignificant) burden.

With this high overlap states truncation scheme in place, we then explore how to *preferentially generate* high overlap states for the truncation scheme. This is discussed in Sec. 3.5 and we illustrate its application in efficiently constructing a given initial state to high precision for a non-perturbative quench. This is not easily achievable within the conventional truncated spectrum approach. We also provide a number of additional convergence checks of our initial state in this section. With this algorithm at hand, we are able to compute real time non-equilibrium dynamics following a quench, as discussed in Sec. 4, and access the long-time limit via the diagonal ensemble.

In Sec. 5 we study strongly non-perturbative quenches, where numerical renormalization group approaches within the high overlap truncation scheme need some modification. A modified algorithm, the matrix element renormalization group (MERG), is detailed in this section. We illustrate problems of the HOSTS algorithm and the success of MERG in computing non-equilibrium dynamics following strongly non-perturbative quenches. Furthermore, we introduce a general version of the matrix element renormalization group algorithm able to deal with excited states as well as ground states. We conclude in Sec. 6, where we also suggest a number of future directions for studies.

## 2 The Lieb-Liniger model

The Lieb-Liniger model describes indistinguishable bosons confined to move in a single spatial dimension, which are coupled via an ultra-local density-density interaction. On a ring of circumference $R$ the Hamiltonian reads [22, 23]

$$H(c) = \int_0^R dx \left( \frac{\hbar^2}{2m} \partial_x \Psi^\dagger(x) \partial_x \Psi(x) + c \Psi^\dagger(x) \Psi^\dagger(x) \Psi(x) \Psi(x) \right). \tag{1}$$

Here $m$ is the mass of boson and $c$ is the interaction strength. Here we will focus on the case of repulsive interactions, $c > 0$, and henceforth we set $2m = \hbar = 1$ to define our units. We will consider the case of unit density $N/R = 1$ herein.

### 2.1 Bethe Ansatz Solution

The Lieb-Liniger model is integrable and exactly solvable [22–24]; $N$-particle eigenstates $|\{\lambda\}_N\rangle$ are characterized by a set of $N$ real rapidities $\{\lambda\}_N = \{\lambda_1, \dots, \lambda_N\}$ that satisfy the Bethe equations

$$e^{-i\lambda_j R} = \prod_{\substack{l=1 \\ l \neq j}}^N \frac{\lambda_l - \lambda_j + ic}{\lambda_l - \lambda_j - ic}. \tag{2}$$

These states have momentum $P(\{\lambda\}_N)$ and energy $E(\{\lambda\}_N)$ given by

$$P(\{\lambda\}_N) = \sum_{j=1}^N \lambda_j, \qquad E(\{\lambda\}_N) = \sum_{j=1}^N \lambda_j^2. \tag{3}$$

Integrability of the model is realized through an infinite family of conserved quantities, whose eigenvalues take the form

$$Q_n(\{\lambda\}_N) = \sum_{j=1}^{N} \lambda_j^n, \quad n = 1, 2, \ldots, \tag{4}$$

where $Q_1 = P$, $Q_2 = E$. We work with eigenstates $|\{\lambda\}_N\rangle$ that are normalized as [24, 70]:

$$\langle \{\lambda\}_N | \{\lambda\}_N \rangle = c^N \prod_{j<l} \frac{(\lambda_j - \lambda_l)^2 + c^2}{(\lambda_j - \lambda_l)^2} \det \mathcal{N}, \tag{5}$$

where $\mathcal{N}$ is the $N \times N$ "Gaudin matrix", with elements

$$\mathcal{N}_{jl} = \delta_{jl}\left(R + \sum_{k=1}^{N} K(\lambda_j, \lambda_k)\right) - K(\lambda_j, \lambda_l) \tag{6}$$

and

$$K(\lambda, \mu) = \frac{2c}{c^2 + (\lambda - \mu)^2}. \tag{7}$$

### 2.1.1 Characterizing eigenstates via integers: Logarithmic Bethe equations

The $N$-particle eigenstates, $|\{\lambda\}_N\rangle$, can be characterized via sets of unique sets of quantum numbers, $\{I\}$, which are integer or half-odd integer (depending on the parity of the particle number $N$). There is a one-to-one correspondence between sets of quantum number and sets of rapidities, defined via the Logarithmic Bethe equations

$$\lambda_j R = 2\pi I_j - 2\sum_{l=1}^{N} \arctan\left(\frac{\lambda_j - \lambda_l}{c}\right), \tag{8}$$

where the quantum numbers satisfy

$$I_j \in \begin{cases} \mathbb{Z} + \frac{1}{2} & \text{for } N \text{ even}, \\ \mathbb{Z} & \text{for } N \text{ odd}, \end{cases} \tag{9}$$

and a Pauli principle, $I_j \neq I_l$ for $j \neq l$.

The mapping between quantum numbers and rapidities satisfies $\lambda_j > \lambda_l$ if $I_j > I_l$ (due to the monotonic nature of the second term on the right of Eq. (8)). From the definition of the energy in terms of the rapidities, Eq. (3), it follows then that the ground state configuration of quantum numbers is a "Fermi sea" of quantum numbers that are symmetrically distributed about the origin. In the large $c$ limit, the rapidities crystallize on to $\lambda_j \to (2\pi/R)I_j$, as if noninteracting fermions.

### 2.1.2 Equilibrium and non-equilibrium properties

The Lieb-Liniger model is perhaps the simplest non-trivial integrable model, with many of its equilibrium properties being well understood. This includes both thermodynamic properties and correlation functions of local operators [24, 71–73]. There are also known expressions for scalar products [74], as well as determinant representations of matrix elements of local operators in the eigenbasis [74–79], some of which are detailed in the appendix (and will be used further in this work). Recently, exact results for the full counting statistics and local correlation functions have been obtained [54].

Non-equilibrium properties of the model following a quantum quench are much less well understood, with important studies only emerging over the past six years [34–40, 42, 43, 47, 48]. Such studies have been rather restricted, relying on knowledge of the *overlaps* of eigenstates of the Hamiltonian at different interaction strengths. These can, in some special limits, be extracted from integrability of the model and simplifications which occur in those limits. Away from these cases, such studies of non-equilibrium properties are hampered by lack of knowledge of the overlaps and a dearth of techniques for calculating them.

Recently, there have been a number of works that study the emergence of non-equilibrium steady states in the Lieb-Liniger, in the context of dynamics starting from inhomogeneous initial states [44, 49, 50, 52–58]. These studies have been enabled by the generalized hydrodynamics framework [44, 80], an adaptation of hydrodynamics to the case of integrable systems. This framework has also allowed the computation of the Drude weight in the Lieb-Liniger model [81]. To be clear, we will be considering only cases with translational invariance here, i.e. global quantum quenches.

## 2.2 The Quench Protocol

We consider the following problem. The system is initialized in the ground state of the Lieb-Liniger model (1) at interaction strength $c_i > 0$. At time $t = 0$ the interaction strength is instantaneously changed $c_i \to c_f > 0$ and the system subsequently evolves in time according to $H(c_f)$. Of interest to us is how to compute the time evolution and long-time limit of expectation values of observables for *generic values* of the initial and final interaction strengths, $c_i$ and $c_f$.

For approaches such as the quench action [5, 19] a crucial role is played by *the overlaps*. The overlaps describe how an initial state $|\Psi_i\rangle$ is projected onto the eigenstates $|\{\lambda\}_N^{(n)}\rangle$ of $H(c_f)$, the Hamiltonian governing time evolution

$$|\Psi_i\rangle = \sum_{n=0}^{\infty} |\{\lambda\}_N^{(n)}\rangle \underbrace{\langle\{\lambda\}_N^{(n)}|\Psi_i\rangle}_{\text{the overlaps}}. \tag{10}$$

Thus the overlaps directly determine how the initial state evolves in time

$$|\Psi_i(t)\rangle \equiv e^{-iH(c_f)t}|\Psi_i\rangle = \sum_{n=0}^{\infty} e^{-iE(\{\lambda\}_N^{(n)})t}|\{\lambda\}_N^{(n)}\rangle\langle\{\lambda\}_N^{(n)}|\Psi_i\rangle. \tag{11}$$

Analytically computing the overlaps is a formidable task, even with the toolbox of integrability at hand. Indeed, it is generally not known how to perform such a calculation, with analytical overlaps having only been obtained in a handful of tractable cases [37, 82–97].

### 2.2.1 Formulation in terms of a perturbed Hamiltonian

The time evolved state, Eq. 11, requires knowledge of how the initial state (the ground state of $H(c_i)$, the initial Hamiltonian) is expressed in terms of eigenstates of the final Hamiltonian, $H(c_f)$. If we can construct the initial Hamiltonian directly in the basis of eigenstates of the final Hamiltonian, diagonalization would yield the overlaps directly. In practice, we are dealing with a continuum bosonic model, so one must truncate the constructed Hamiltonian to obtain a finite matrix that one can diagonalize. This is a so-called truncated spectrum approach or approximation.

The manner in which we are formulating this problem, working directly with strongly correlated basis states, is different to previous applications of truncated spectrum approaches to non-equilibrium dynamics [61–68]. In these cases, a computational basis of *non-interacting*

*fermions/bosons* was used, with both the initial state and final eigenbasis being constructed from these computational states.

For the case at hand, we are able to construct the initial Hamiltonian in the final basis through exact knowledge of eigenstates and matrix elements from integrability of the model [78, 79]. We begin by writing the Hamiltonian in the form

$$H(c_i) = H(c_f) + (c_i - c_f) \int_0^R dx\, \Psi^\dagger(x)\Psi^\dagger(x)\Psi(x)\Psi(x). \tag{12}$$

In this manner, we have written the initial Hamiltonian as a 'perturbation' of the final Hamiltonian.[2] In the ground state (zero momentum) sector with fixed particle number $N$, the matrix elements of the initial Hamiltonian can then be written as

$$\langle \{\lambda\}_N^{(m)} | H(c_i) | \{\lambda\}_N^{(n)} \rangle = \delta_{n,m} E\big(\{\lambda\}_N^{(n)}\big) + (c_i - c_f) R \langle \{\lambda\}_N^{(m)} | \big(\Psi^\dagger(0)\big)^2 \big(\Psi(0)\big)^2 | \{\lambda\}_N^{(n)} \rangle. \tag{13}$$

Here, as above, $|\{\lambda\}_N^{(n)}\rangle$ are $N$ particle eigenstates of the *final Hamiltonian $H(c_f)$*. We will often call these "computational basis states". We see from (13) that we require matrix elements of the operator $g_2(0) = \big(\Psi^\dagger(0)\big)^2 \big(\Psi(0)\big)^2$ between computational basis states. Known results for these are recapitulated in Appendix A.

## 3 Developing a high overlap states truncation scheme

### 3.1 The ideal truncation scheme

At the heart of the problem under study is how to truncate the initial Hamiltonian, constructed in the computational basis, to obtain optimal convergence of physical quantities. The time evolved wave function (11) clearly points the way. Consider organizing the computational basis by the magnitude of the overlap $w^{(n)} = |\langle \{\lambda\}^{(n)} | \Psi_i \rangle|$. Truncation to the $N_{\text{tot}}$ computational states with highest overlaps

$$|\Psi(t)\rangle_{\text{approx}} = \sum_{n=0}^{N_{\text{tot}}} e^{-iE(\{\lambda\}^{(n)})t} |\{\lambda\}^{(n)}\rangle \langle \{\lambda\}^{(n)} | \Psi_i \rangle, \tag{14}$$

will give bounded errors for (bounded) physical observables. That is, saturating the norm of the state $|\Psi_i\rangle$ to

$$s(N_{tot}) = 1 - \sum_{n=0}^{N_{\text{tot}}} \big| \langle \{\lambda\}^{(n)} | \Psi_i \rangle \big|^2, \tag{15}$$

the maximal error $\epsilon_{\max}[\cdot]$ on the time evolution of a bounded operator $A$ is

$$\epsilon_{\max}\big[ A(t) - A(t)_{\text{approx}} \big] = s(N_{\text{tot}}) \max_{m,n}(A_{m,n}), \tag{16}$$

Here $A(t)_{\text{approx}}$ is the operator evaluated within the time evolved approximate state $|\Psi(t)\rangle_{\text{approx}}$, $A_{m,n} = \langle \{\lambda\}^{(m)} | A | \{\lambda\}^{(n)} \rangle$ are the matrix element of operator $A$ in the computational basis, and $\max_{m,n}(A_{m,n})$ denotes its maximal value. Thus if $s(N_{\text{tot}})$ is sufficiently small, for any bounded operator the errors are small.

An expansion such as Eq. (14) is all well and good, but we do not *a priori* know the overlaps. Thus we are unable to order the computational basis according to the overlaps, and we must

---

[2]For the approach that is being discussed, the strength of this 'perturbation' $(c_i - c_f)$ does not need to be small. This will be explicitly demonstrated in the results that follow.

develop an approach that mimics this. This is the subject of the remainder of this section, where we develop a "high overlap states truncation scheme". We do so in a sequence of steps, drawing inspiration from conventional truncated spectrum methods, adaptations, and their failures, to eventually arrive at an efficient high overlap states truncation scheme.

## 3.2 The truncated spectrum approach

The Lieb-Liniger model is a continuum field theory of interacting bosons. The Hilbert space is spanned by infinitely many states, and the Hamiltonian is thus a matrix with infinite dimensions. In the formulation of our problem as a perturbed Hamiltonian, Sec. 2.2.1, the perturbed Hamiltonian is a dense matrix in the computational basis. To proceed, we have to truncate the Hilbert space in some manner to obtain a finite matrix, which can then be diagonalized to obtain the eigenstates (and their energies) and hence the overlaps.

As a starting point, we take inspiration from standard truncated spectrum methods [60, 98, 99]. If the perturbing operator in Eq. (12) is renormalization group relevant, it will:

1. Flow to strong coupling as the renormalization group is taken to the low energy limit, leading to a strong mixing between low-energy states in the computational basis $|\{\lambda\}^{(n)}\rangle$.

2. Flow to weak coupling in the ultraviolet (high energy), meaning that high energy states $|\{\lambda\}^{(n)}\rangle$ are approximate eigenstates of the perturbed Hamiltonian too.

3. As a corollary to the above points, the operator cannot strongly couple low-energy and high-energy states in the computational basis.

In our scenario, in the non-interacting limit the perturbing operator has scaling dimension 'zero'.[3] This is similar to the scenario encountered in the $1 + 1$D $\phi^4$ theory, which has been studied extensively with truncated spectrum methods [101–111]. This suggests that perhaps the same method may achieve success here.

The simplest possible truncation, motivated by the 'decoupling' of low- and high-energy computational basis states, is to introduce an energy cutoff $\Lambda$ and consider all computational states with energy below the cutoff. This is the truncation originally envisaged by Yurov and Zamolodchikov in the context of perturbed conformal field theories [98, 99]. Convergence of the ground state energy (for example) can then be checked as a function of the cutoff energy $\Lambda$. As a first example, we show an example of this for the $c_i = 20$ ground state of ten particles (constructed in terms of $c_f = 10$ computational basis states) in Fig. 1. The convergence of the ground state energy with $\Lambda$ is consistent with an exponential fit (although, we note, that we do not have many decades of data to fit over).

For many models (see the review article [60]) it has been found that convergence can be slow, requiring energy cutoffs far beyond those one can treat with exact diagonalization. This can be seen clearly in Fig. 1: to get the ground state energy to within just 1% of the exact value, we would expect to have to include many hundreds of thousands of states. Various techniques have been developed to counter this, as discussed in [60], ameliorating the effects of the cutoff. In the following section we discuss and implement one such approach: a numerical renormalization group extension.

## 3.3 Numerical renormalization group extension

To combat slow convergence of the eigenstates and eigenvalues, we supplement the truncated spectrum procedure with a numerical renormalization group extension. The numerical renor-

---

[3]That is, the two-point function of the free bosonic field is logarithmic in form. In the conformal field theory context [100], this reflects the fact that $\Psi$ is not a primary field. See, e.g., Ref. [101] for a detailed discussion of the analogous case in the scalar $\phi^4$ model.

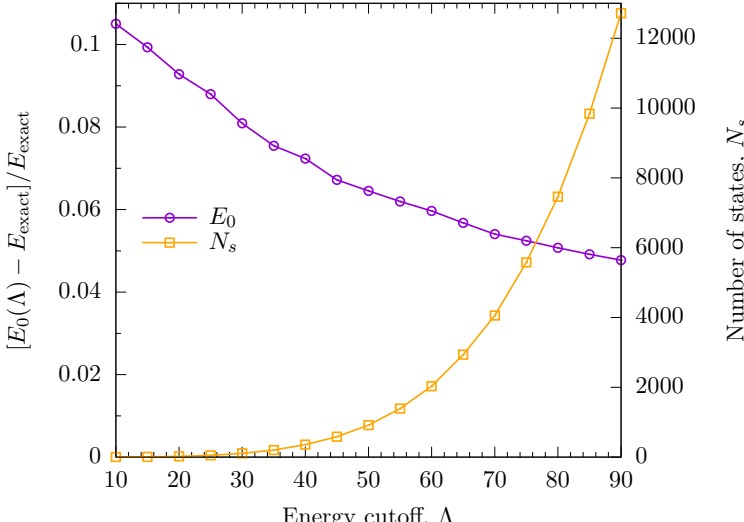

Figure 1: The ground state energy $E_0$ (compared to the exact result $E_{\text{exact}}$) and the number of basis states $N_s$ as a function of the energy cutoff $\Lambda$. The ground state of the Hamiltonian (1) with $c_i = 20$ is constructed in terms of eigenstates for $c_f = 10$, with $N = 10$ particles at unit density, using the truncated spectrum approach.

malization group was first introduced by Wilson to tackle the Kondo problem [112] and since then has become a vital tool for tackling impurity problems [113], including in the context of dynamical mean field theory (see, e.g., Ref. [114]). Its application to truncated spectrum methods was first suggested by Konik and Adamov in 2007 [115], and has since been applied to tackle a number of problems beyond the reach of the plain truncated spectrum approach [116–121].

The numerical renormalization group procedure for the truncated spectrum approach is formulated as follows:

1. Construct the computational basis $\{|\{\lambda\}^{(j)}\rangle\}$ and order by energy $E(\{\lambda\}^{(j)})$.

2. Construct a truncated Hamiltonian from the first $N_s + \Delta N_s$ computational basis states, $\left\{|\{\lambda\}^{(1)}\rangle, \ldots, |\{\lambda\}^{(N_s+\Delta N_s)}\rangle\right\}$ and diagonalize it to obtain approximate energies and eigenstates, $\left\{|E^{(1)}\rangle, \ldots, |E^{(N_s+\Delta N_s)}\rangle\right\}$.

3. Discard the highest $\Delta N_s$ approximate eigenstates $\left\{|E^{(N_s+1)}\rangle, \ldots, |E^{(N_s+\Delta N_s)}\rangle\right\}$, from the truncated Hamiltonian.

4. Construct a new basis of $N_s + \Delta N_s$ from the remaining $N_s$ approximate eigenstates and the next $\Delta N_s$ states in the computational basis.

5. Construct the Hamiltonian in this new basis, and diagonalize it to obtain new approximations to the eigenstates and their energies.

6. Return to the third step.

This process is continued, obtaining new approximate eigenstates after each cycle of steps 3 to 5, until the required convergence of the ground state energy/eigenstate is reached or the computational basis is exhausted. With such a procedure, it is possible to construct the ground state of a perturbed Hamiltonian in terms of many hundreds of thousands or millions of the computational basis states [116, 122].

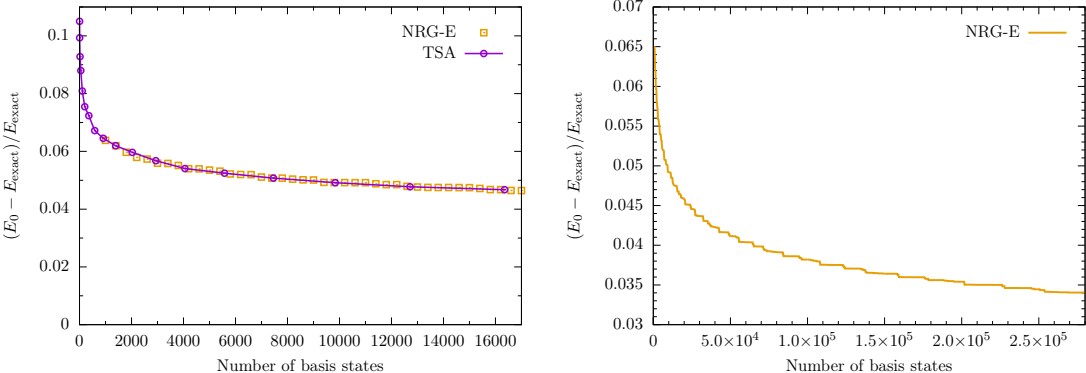

Figure 2: Convergence of the approximate ground state energy $E_0$ with the number of basis states, as computed using the truncated spectrum procedure (TSA) and its numerical renormalization group extension (NRG-E). The $c_i = 20$ ground state of the Hamiltonian (1) is constructed in terms of $c_f = 10$ eigenstates for 10 particles. NRG-E is performed with $N_s = 600$ and $\Delta N_s = 200$ (corresponding to an initial energy cutoff of $\Lambda \approx 50$). We plot every other NRG-E step; excellent agreement between the results of the numerical renormalization group and the truncated spectrum approach is seen (left panel). The numerical renormalization group procedure can access number of basis states far beyond those accessible to full diagonalization (right panel).

As an illustration, in Fig. 2 we present the convergence of the ground state energy $E_0$ with $c_i = 20$ as a function of the number of computational basis states considered in the numerical renormalization group procedure. The computational basis is formed from eigenstates of the Hamiltonian with $c_f = 10$. As a first check, the numerical renormalization group procedure (performed with $N_s = 600$ and $\Delta N_s = 200$, corresponding to an energy cutoff at the first step of the procedure of $\Lambda \approx 50$) is compared to full diagonalization in Fig. 2(a). Despite the small size of the numerical renormalization group Hamiltonian (of total dimension $N_s + \Delta N_s = 800$), we see that the obtained results accurately reproduce the full truncated spectrum results of Fig. 1. The numerical renormalization group does, however, allow us to consider many more basis states than can be tackled with with full exact diagonalization in a time and memory efficient manner. This is illustrated in Fig. 2(b), where we consider 280,000 computational basis states in our numerical renormalization group procedure. This allows us to converge energies to below 3.5%[4] at the end of the procedure, which is significantly smaller than the level spacing $E_1 - E_0$ for the parameters under consideration.

## 3.4 Ordering by an alternative metric

As can be seen in Fig. 2(b), the convergence of the ground state energy $E_0$ with the number of basis states is slow: To reach a precision of 2% it is likely that one will need to consider more than $10^6$ basis states. It is also evidence that the convergence obtained within the numerical renormalization group procedure has a lot of structure: there are steps of the procedure where the ground state energy is approximately constant, whilst at other steps it rapidly drops. One can then ask: is there an alternative ordering of the computational basis states that prioritizes the "important" states, where these large drops occur, and so improve the convergence?

To begin tackling this problem of modifying the ordering metric, we follow the suggestions

---

[4]This corresponds to approximately 11% with respect to the Fermi energy.

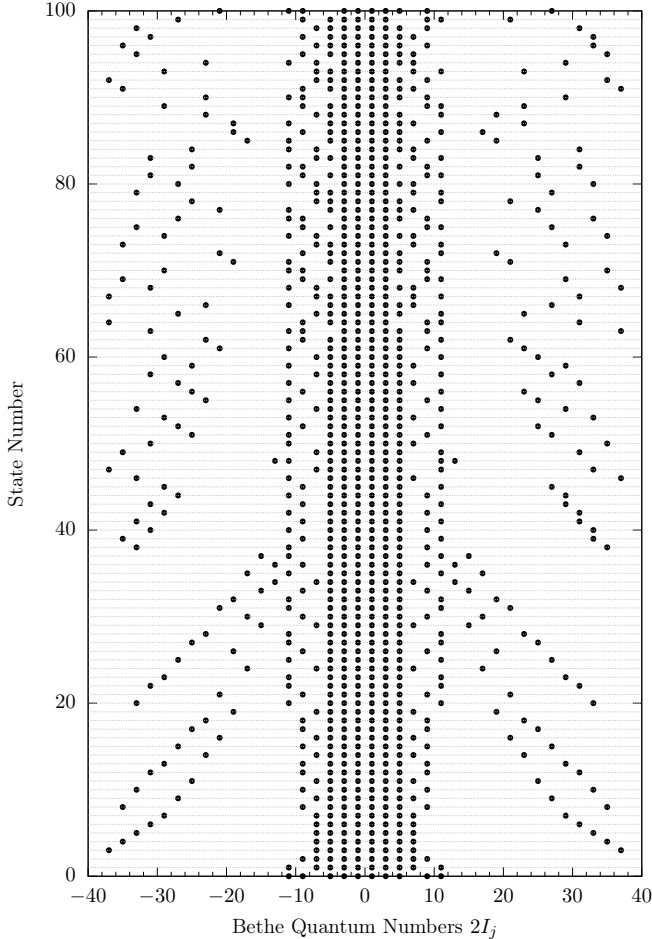

Figure 3: The configurations of quantum numbers $\{I_j\}$, see Eq. (8), characterizing the first 100 basis states ordered via the matrix element metric, Eq. (17) for the $c_i = 20$ to $c_f = 10$ quench with $N = 10$ particles. The total basis was formed from 273,358 states generated by the ABACUS scanning routine. The highest weight states, according to the metric, have lowest 'state number'. Note that some of the highest weight states contain high momentum (large quantum number $2I_j$) excitations.

of Refs. [18, 119, 122, 123] (see also the discussion in [60]) and take a pragmatic approach. We order the computational basis states according to the values of the matrix elements

$$\left| \langle \{\lambda\}_N^{(n)} | g_2(0) | \tilde{E}_j \rangle \right|, \quad j = 0, 1, 2. \tag{17}$$

Here $|\tilde{E}_j\rangle$ are the three lowest energy eigenstates of the initial Hamiltonian, i.e. those states we are trying to construct. In practice, $|\tilde{E}_j\rangle$ are first constructed via the truncated spectrum approach with a small energy cutoff (corresponding to circa two thousand states) and these approximate eigenstates are then used to construct the matrix elements (17). This procedure attempts to capture those states $|\{\lambda\}_N^{(n)}\rangle$ that hybridize with and contribute most strongly to the low energy states.

To get some understanding of how this change of metric, Eq. (17), modifies the states being considered within the numerical renormalization group procedure, we present the configurations of quantum numbers $\{I_j\}$ (recall Eq. (9)) that characterize the one hundred highest weight states according to this metric. We show these in Fig. 3. There is clearly a significant

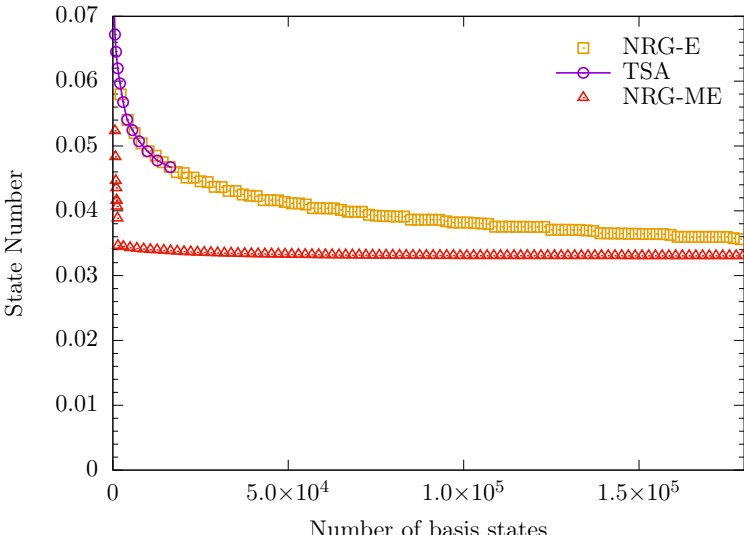

Figure 4: A comparison between the numerical renormalization group (NRG-E) results of Fig. 2(b) and the modified numerical renormalization group with matrix element ordering (NRG-ME). We see that the alternative ordering leads to massive improvement in the convergence of the ground state energy (and similar improvement is seen in low-lying excited states) for fixed number of basis states. NRG-ME was performed with $N_s = 600$, $\Delta N_s = 120$ with a total of $280,000$ basis states.

change in ordering of the states as compared to energy ordering. Most of the highest weight states under the metric (17) describe pairs of "highly excited quantum numbers" that have moved away from the "Fermi sea" of quantum numbers centered on zero, leaving behind holes. We see that in a family of states with fixed configuration of quantum numbers close to zero, states containing the most excited quantum numbers generally have highest weight. It is also apparent that one of the highest weight states is the ground state of the final Hamiltonian.

### 3.4.1 Convergence of the ground state energy with matrix element metric

The reordering presented in Fig. 3 seems a little surprising, but leads to considerable improvement in the convergence of the numerical renormalization group results. This is shown in Fig. 4, where after only seven numerical renormalization group steps, the convergence of the ground state energy is already lower than that obtained with over $10^5$ steps of the energy-ordered numerical renormalization group procedure. For the computational basis considered, we have essentially saturated our approximate representation of the initial state. With this significant improvement in convergence, we work with alternative (non-energy ordered) metrics in the remainder of this work.

The problem with the procedure as laid out, at the moment, is there is still a need to generate a very large computational basis, compute the weight according to the metric (17), reorder, and then perform the numerical renormalization group procedure. The total size of this computational basis essentially introduces an energy cutoff and this limits the extent to which we can saturate the approximate representation of the state.

Crucially, insights from the following two sections will allow us, in Sec. 3.5, to throw off the shackles of needing to generate a large computational basis to matrix element order, and instead we will realize a way to *preferentially generate the high overlap states*.

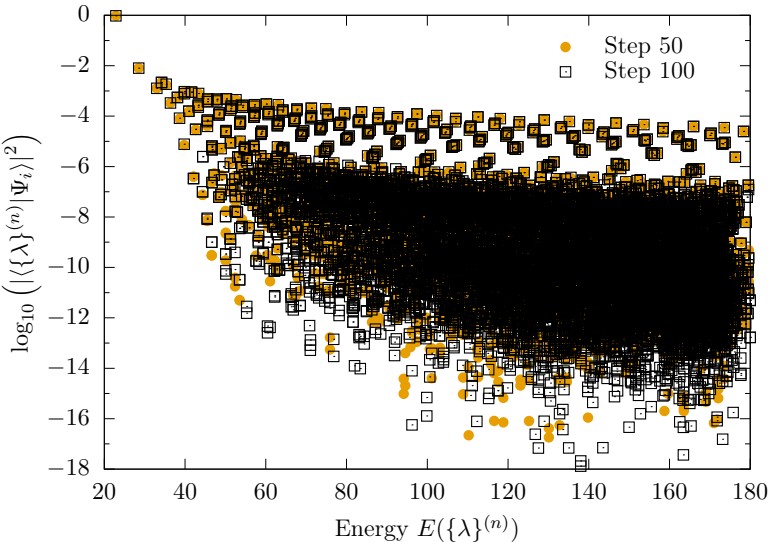

Figure 5: The overlaps $\langle \{\lambda\}^{(n)}|\Psi_i\rangle$ between the ground state of the Lieb-Liniger model with $c_i = 20$ and eigenstates $|\{\lambda\}^{(n)}\rangle$ of the $c_f = 10$ Lieb-Liniger model with energy $E(\{\lambda\}^{(n)})$. The ground state $|\Psi_i\rangle$ is constructed in terms of the NRG-ME procedure, and we present results after 50 and 100 steps, showing that the dominant overlaps are well-converged. This is not surprising in light of Fig. 4.

### 3.4.2 The overlaps: Convergence and structure

Beyond examining the convergence of the energy, other checks are critical in ascertaining the validity of results obtained within the truncated spectrum and its numerical renormalization group extensions. Above, we are able to compute exactly (from the Bethe ansatz) the energy to which our obtained state should be approaching, giving us a quantitative measure of convergence. We have seen that convergence can be obtained provided a sufficiently large number of eigenstates are included in the computational basis.

Having constructed an approximation to our initial state, we directly have the overlaps at hand. These are essential for computing non-equilibrium dynamics and the long time steady state following a quench. It is worthwhile, at this point, to examine that the overlaps themselves are well converged (which should follow directly from the convergence of the energy). In Fig. 5 we plot the square of the overlaps $|\langle\Psi_i|\{\lambda\}^{(n)}\rangle|^2$ as a function of the energy $E(\{\lambda\}^{(n)})$ of the computational basis state $|\{\lambda\}^{(n)}\rangle$.[5] Results are presented at two well-separated steps of the numerical renormalization group procedure (conducted with metric (17)). We see clearly that large overlap computational basis states large have well-converged overlaps, being almost identical at the two different steps of the procedure. The computational states with very small overlaps are physically unimportant (recall Sec. 3.1) and clearly subject to floating point errors of the numerical implementation.

A significant message to take from Fig. 5 (and implicitly from Fig. 4) is that computational basis states with large energies can contribute significantly to the initial state. Indeed, in

---

[5]We note that these overlaps are related to the statistics of work done [124]:

$$P(W) = \sum_n \delta\left(W - E(\{\lambda\}_N^{(n)})\right)\left|\langle\Psi_i|\{\lambda\}_N^{(n)}\rangle\right|^2,$$

giving the coefficients of the delta functions. This quantity has been studied, for example, in the Ising field theory in Refs. [61, 63].

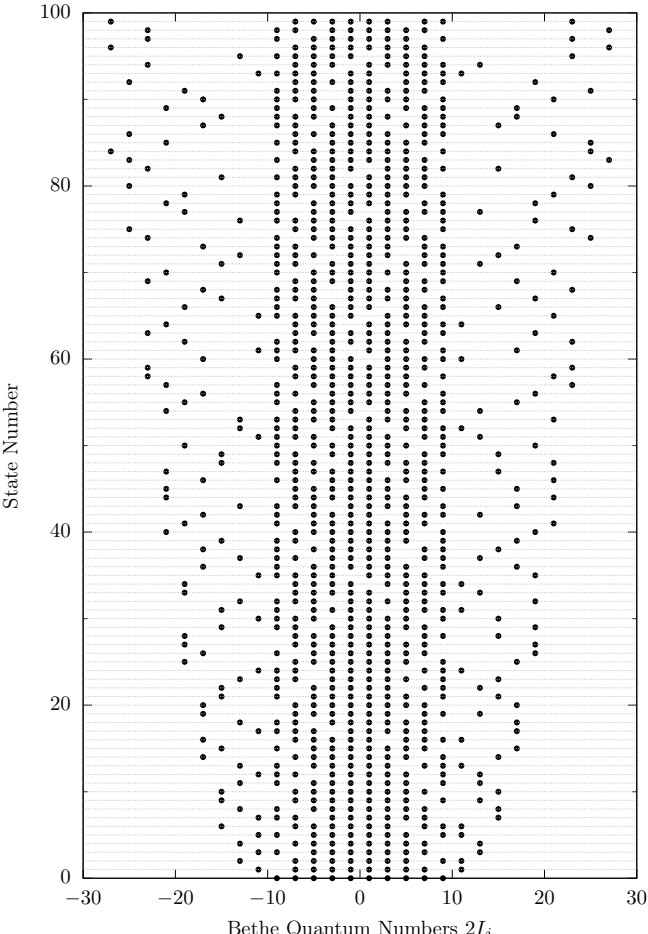

Figure 6: The configurations of quantum numbers $\{I_j\}$, see Eq. (9), characterizing the 100 computational basis states with highest overlaps in the $c_i = 20$ ground state constructed in terms of $c_f = 10$ eigenstates. These were obtained from a numerical renormalization group procedure with the basis ordered according to the metric (17). States with larger overlaps appear towards the bottom of the figure.

Fig. 5 we see a band of "high overlap states" with square overlaps $\sim 10^{-4} - 10^{-5}$ extending out to high energies. We note that the results of Fig. 5 are well-converged with numerical renormalization group step, but cannot be well-converged with regards to increasing the size of the computational basis. This is evident from the fact that our procedure produces states with $\langle \Psi_i | \Psi_i \rangle_{\text{approx}} = 1$, and there is no reason to believe high overlap states stop at energy $E(\{\lambda\}^{(n)}) = 180$, the effective energy cutoff of our computational basis. Note that this does not imply that physical quantities are not well converged with the size of the basis (we indeed observe that physical quantities *are* well converged).

With convergence at fixed computational basis size confirmed for the high overlap states, let us now illustrate the structure of these high overlap states. From the point of view of analytical calculations, there are only a handful of examples where overlaps can be computed [37, 42, 43, 84, 85, 125], so numerical routines have significant potential when brought to bear on such a problem.

For the $c_i = 20 \rightarrow c_f = 10$ quench, the configurations of the quantum numbers in the highest 100 overlap states are shown in Fig. 6. The states are organized from highest weight (bottom of the plot) to lowest weight (top of the plot). We see that the highest weight overlap

is with the ground state of the final Hamiltonian in this case. As we proceed up the plot, we see a pair of excited integers move away from the Fermi sea about the origin, and the holes left behind moving around within the Fermi sea.

At first glance, it may be a little surprising that the high overlap state integers shown in Fig. 6 are ordered so differently to those of matrix element metric, Fig. 3. In some sense, this tells us that we are dealing with an "easy quench" where the overlaps rapidly converge converge if we get approximately the correct ordering metric. In the next section we will construct an alternative metric, taking some inspiration from the information in Fig. 6 and combining it with other knowledge, that more accurately reproduces the optimal ordering.

### 3.5 Efficient generation of high overlap states

In the previous sections we have established the efficacy of ordering the computational basis based upon information about the perturbing operator (recall Sec. 2.2.1). However, as we have already highlighted, there is a clear issue with the procedure that has been discussed. This is best illustrated by Figs. 4 and 5: The saturation of the error in the ground state energy in this modified numerical renormalization group procedure is ultimately set by the energy cutoff of the truncated basis on which we perform the new ordering. The behavior of the overlaps as a function of energy, Fig. 5, clearly shows that high energy computational basis states can contribute significantly to the initial state. Indeed, we see in Fig. 5 that computational basis states with $E \sim 10^2 - 10^3$ can have square overlaps as large as $\sim 10^{-4}$.

Following this procedure, if we wish to achieve precision of sub-1% in the energy of the state for ten particles, we will need to first generate a large basis of size $\sim 10^6$, then order according to the metric (17), and then perform a truncated spectrum or numerical renormalization group procedure. The ordering step requires computing matrix elements for *all computational basis states*. This is computationally costly, even if much more efficient than working with an energy-ordered numerical renormalization group procedure. Instead, it would be much better if we could preferentially generate the high overlap states necessary for our algorithms.

In this section we will formulate such a preferential state generation procedure. This will be based upon the philosophy of the ABACUS (**A**lgebraic **B**ethe **A**nsatz-based **C**omputation of **U**niversal **S**tructure factors) Hilbert space scanning algorithm. A general overview of this approach, developed to tackle the computation of equilibrium dynamical correlation functions, can be found in Ref. [126].

The essential insight for applying ABACUS-inspired methods to the non-equilibrium problem at hand is contained within Figs. 6. There one can see that the largest overlap is with the ground state of the final Hamiltonian, herein denoted $|\{\lambda\}^{(0)}\rangle$. The matrix element metric, Eq. (17), then approximately orders the states that hybridize most strongly with $|\{\lambda\}^{(0)}\rangle$ via the perturbing operator $g_2(x)$, i.e. the $|\{\lambda\}^{(n)}\rangle$ that maximize

$$\int dx \, \langle\{\lambda\}^{(0)}|g_2(x)|\{\lambda\}^{(n)}\rangle \propto \delta(P_0 - P_n) \langle\{\lambda\}^{(0)}|g_2(0)|\{\lambda\}^{(n)}\rangle. \tag{18}$$

Here we use the short hand $P_n = P(\{\lambda\}^{(n)})$. Formulated in this manner, the case for applying an ABACUS-like algorithm is clear. Consider computing the *equilibrium* dynamical correlation function of $g_2(x)$:

$$S_{g_2}(k, \omega) \propto \int_{-\infty}^{\infty} \int_{-\infty}^{\infty} dx \, dt \, e^{i(kx+\omega t)} \langle\{\lambda\}^{(0)}|g_2(x,t)g_2(0)|\{\lambda\}^{(0)}\rangle. \tag{19}$$

Here $g_2(x,t) = e^{iHt}g_2(x)e^{-iHt}$ is the time evolved $g_2$ operator. The dynamical correlation function $S(k, \omega)$ can be evaluated by inserting the resolution of identity between the two

operator and summing the resulting Lehmann spectral representation

$$S_{g_2}(k,\omega) \propto \sum_n \delta(\omega - E_n + E_0)\delta(k - P_n + P_0)\left|\langle\{\lambda\}^{(0)}|g_2(0)|\{\lambda\}^{(n)}\rangle\right|^2. \tag{20}$$

Once again, we use a short hand notation $E_n = E(\{\lambda\}^{(n)})$. Thus the states with highest weight under the metric (17) are (approximately) those that contribute most strongly to Eq. (20) – the problem that ABACUS was designed to tackle.[6]

We are, of course, not aiming to compute equilibrium correlation functions here, but instead non-equilibrium dynamics. By approximating the initial state in the metric (17) by $|\{\lambda\}^{(0)}\rangle$ we can motivate an ABACUS-like scheme to generate the states with high weight on this metric. We can, however, draw some inspiration from perturbation theory to construct a better metric. If we have the state with the highest overlap with the state we are trying to construct (here this state is $|\{\lambda\}^{(0)}\rangle$), the perturbation theory tells us the first order term in the expansion for the approximation state should be

$$|\Psi_i\rangle_{\text{approx}} = |\{\lambda\}^{(0)}\rangle + (c_i - c_f)R\sum_{m\neq 0}\frac{\langle\{\lambda\}^{(m)}|g_2(0)|\{\lambda\}^{(0)}\rangle}{E_0 - E_m}|\{\lambda\}^{(m)}\rangle + O(g^2). \tag{21}$$

Thus a better metric is obviously apparent: we should organize our computational basis states according to the their weights

$$w\left(|\{\lambda\}^{(n)}\rangle\right) = \left|\frac{\langle\{\lambda\}^{(n)}|g_2(0)|\{\lambda\}^{(0)}\rangle}{E_n - E_0 + \epsilon}\right|. \tag{22}$$

Here $\epsilon$ is a simple numerical factor introduced to avoid a divergence for the case of $(n) = (0)$ (we take $\epsilon = 0.1$). States generated in an ABACUS-like scanning according to their matrix elements weights can easily be post-sorted according to (22).[7]

Implementing this procedure, the 400 highest weight states for the $c_i = 20 \rightarrow c_f = 10$ quench for ten particles are shown in Fig. 7. This is clearly rather different to the pure matrix element ordering, Fig. 3, and is much more in keeping with the results shown in Fig. 6. In the next section we will see that this ordering leads to excellent convergence of the initial state energy, and the ordering may be close to optimal in some scenarios (we will later see an example where this does not appear to be the case). In light of the data presented in Fig. 7, it is hardly surprising that energy-ordering the basis fails to give good convergence: Even within the first 400 state there are states with highly excited quantum numbers, i.e. states with high energies. As we are now able to generate computational basis states without an implicit (or explicit) energy cutoff, we expect to be able to saturate the energy of the initial state to a much larger extent (recall Fig. 4).

We call this procedure a "high overlap states truncation scheme." We note that we used both ABACUS and an independently written Hilbert scanning routine within this manuscript. This has helped provide independent checks of the preferential state generation for all our results.

## 3.6 Checking convergence within the high overlap states truncation scheme

With a high overlap states truncation scheme at hand, there is no longer an energy cutoff within the computational basis. The presence of an energy cutoff ultimately governed the value to

---

[6]If the largest overlap was not a ground state, the same procedure could be implemented with $|\{\lambda\}^{(0)}\rangle$ replaced with the largest overlap state, with obvious modifications to Eqs. (21) and (22).

[7]We note that ABACUS-like scanning routines do not generally produce states contributing to Eq. (20) in a monotonically decreasing manner [126]. Thus post-sorting for a consistent truncation scheme is a necessity in any case.

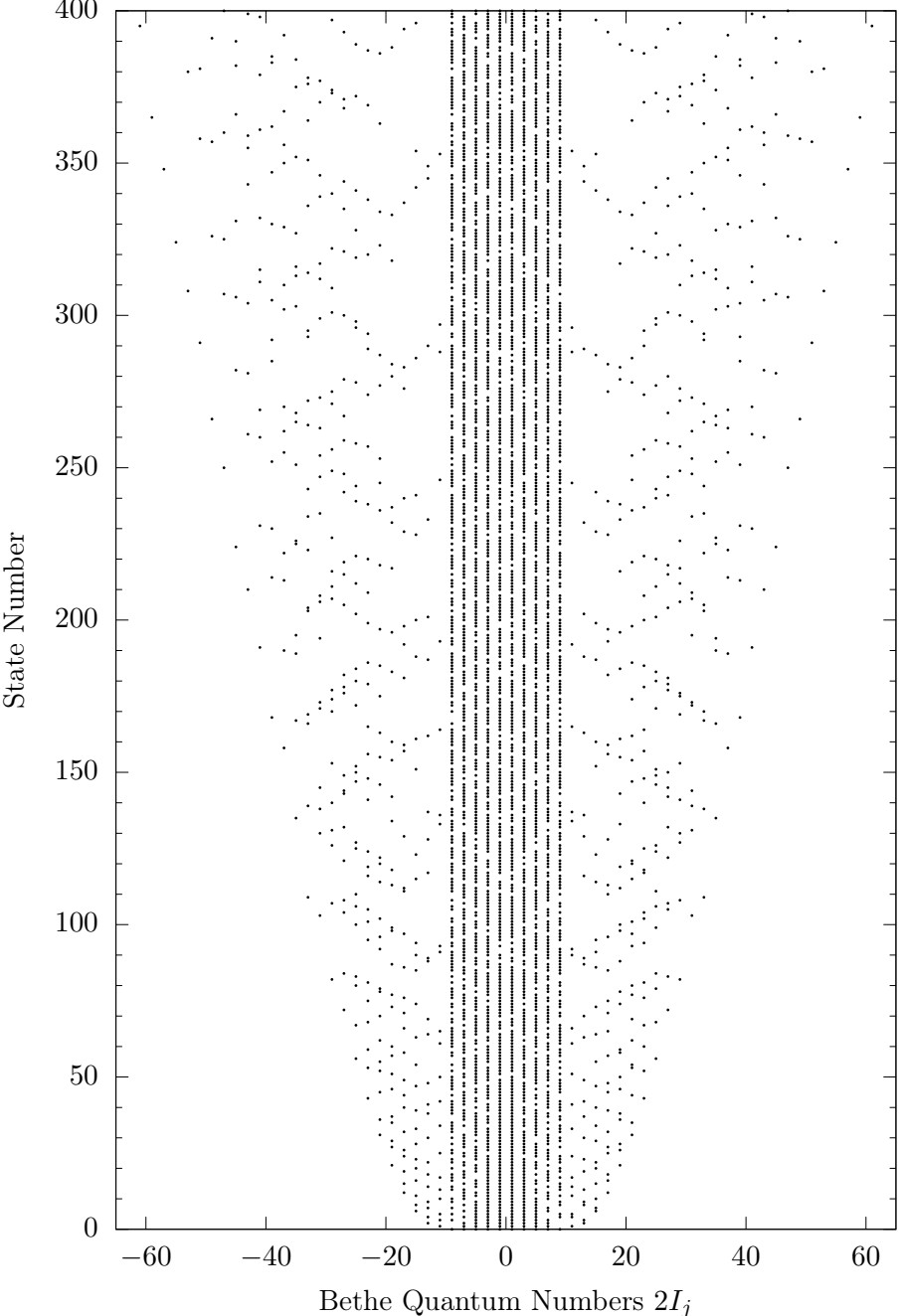

Figure 7: The configurations of integers $\{I_j\}$ in the first 400 states generated via preferential scanning and ordered according to the metric (22) for the $c_i = 20 \rightarrow c_f = 10$ quench. (Highest weights correspond to lowest state numbers.) Notice the similarity with Fig. 6, the output of the NRG-ME procedure; the preferential scanning algorithm efficiently generates those basis states with largest overlaps.

(a)

(b)

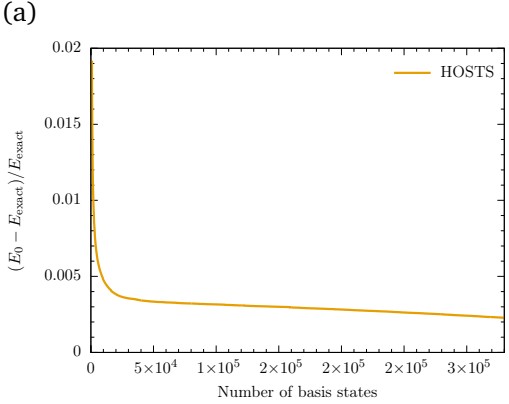

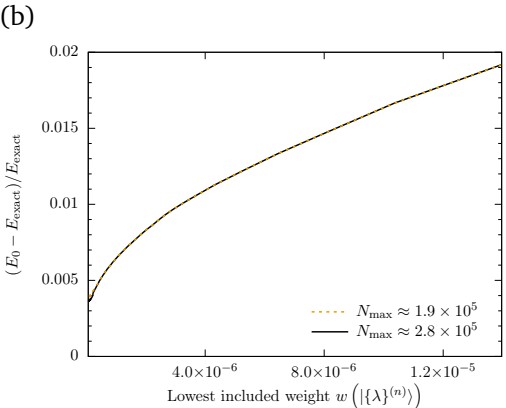

Figure 8: The convergence of the energy $E_0$ of the $c_i = 20$ ground state constructed in terms of $c_f = 10$ eigenstates via the numerical renormalization group within the high overlap states truncation scheme (HOSTS). (a) $E_0$ as a function of number of basis states; $N_{\mathrm{max}} \approx 3.9 \times 10^5$ states are generated via ABACUS and ordered according to their weights (22). (b) $E_0$ as a function of the lowest included weight in the first two hundred steps of the numerical renormalization group procedure for two different total basis sizes, $N_{\mathrm{max}}$. In both (a) and (b) the numerical renormalization group procedure is performed with $N_s + \Delta N_s = 800$ and $\Delta N_s = 160$. Convergence of $E_0$ to under 1% is achieved with only a few thousand basis states, cf. Fig. 4. The convergence w.r.t. the Fermi energy is also smaller than 1% towards the end of the procedure.

which the energy of the initial state could saturate in the previous sections (for example, in Fig. 4 the maximum saturation to within $\sim 0.035 E_{\mathrm{exact}}$). This means that now we can saturate agreement to much less than 1%, while doing so at a significantly decreased computational burden.

Before discussing this in more detail, it is worth first re-evaluating how we assess convergence within the high overlap states truncation scheme. So far, we have checked how the energy of the state varies with the number of computational basis states, but it is not entirely clear how one and if one can extrapolate these results to understand the exact one. There is, for example, no obvious scaling law for the energy as a function of number of basis states.

Within our truncation scheme, a central role is played by the weights of the computational basis states under the metric (22). One potential option for assessing convergence of computed quantities is to plot against this weight (i.e., plotting quantities as a function of the smallest considered weight (22)). We will see that this leads to a reasonable extrapolation scheme, as compared to the number of considered computational basis states.[8]

### 3.6.1 Convergence of the energy

Let us now examine the convergence of the energy of the initial state constructed with the basis of high overlap states via the numerical renormalization group. This is shown, as a function of the number of basis states in Fig. 8(a). The computational basis states are generated preferentially by running the ABACUS algorithm for 30 seconds and then reordering the generated states according to the metric (22). This yields an ordered basis of 220,743 states, on which

---

[8]We note that as we study a continuum model, we cannot guarantee that we generate all states above a given weight of the metric (22), especially when this becomes small. Additional checks, such as convergence of results with the number of states generated and ordered according to the metric (22) must also be performed.

(a)    (b)

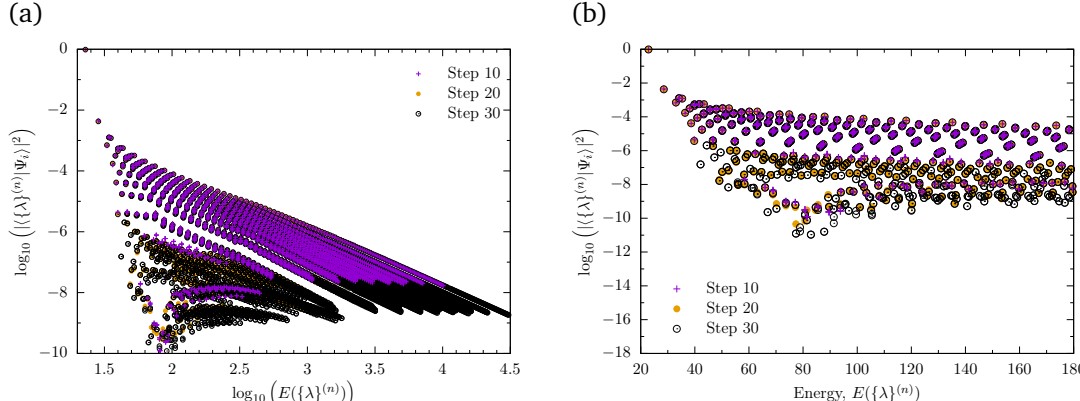

Figure 9: (a) The convergence of the overlaps at different steps of the numerical renormalization group procedure implemented within the high overlap states truncation scheme. The $c_i = 20$ ground state constructed is constructed in terms of $c_f = 10$ eigenstates. Parameters of the procedure are as in Fig. 8. (b) A focused region of the plot, to be compared directly with Fig. 5.

we subsequently perform the numerical renormalization group procedure (in fact, we see that excellent convergence is achieved for basis sizes accessible to full diagonalization). We see very rapid convergence of the initial state energy, requiring only a few thousand computational states to obtain a convergence of under 1%. This should be contrasted to the traditional energy ordering, see Figs. 1 and 2, where we would likely require $> 10^6$ computational basis states to reach the same level of convergence.

In Fig. 8(b) we also present the convergence of the initial state energy $E_0$ as a function of the lowest weight (22) included in each iteration of the numerical renormalization group procedure. We show data for two different total basis sizes $N_{tot}$, which shows that at very small included weights there is some dependence on $N_{tot}$. This implies that our preferential state generation routine has not generated all the computational basis states with weights above a given small value.

### 3.6.2 Convergence of the overlaps

We have just seen that the high overlap states truncation scheme yields excellent convergence of the initial state energy. Let us now turn attention to the overlaps themselves, and how these converge. We present example data in Fig. 9 for the $c_i = 20 \rightarrow c_f = 10$ quench with ten particles. In Fig. 9(a) we present the overlaps (as a function of energy of the computational basis state) at three steps of the numerical renormalization group procedure. From this figure, we can make a few observations.

Firstly, we observe that the high overlap states truncation scheme is indeed preferentially targeting high overlap states. The overlaps generated at early stages of the numerical renormalization group procedure are larger and remain well converged at later steps. Secondly, we see that the quench generates very high energy states: by just the thirtieth step of the procedure, we are probing states with energies $E(\{\lambda\}^{(n)}) \gg 10^4$ (for reference, the ground state energy is $E_{\text{exact}} = 26.9684027\ldots$). Thirdly, there appear to be clear "families of states" within this plot, whose overlaps at high energies can quite easily be predicted by extrapolation.

We can directly compare the results of our computation to those in Fig. 5, obtained from the original matrix element ordering without preferential generation of high overlap states. As compared to Fig. 9(b), we see that the high overlap states truncation scheme avoids dealing

(a)

(b)

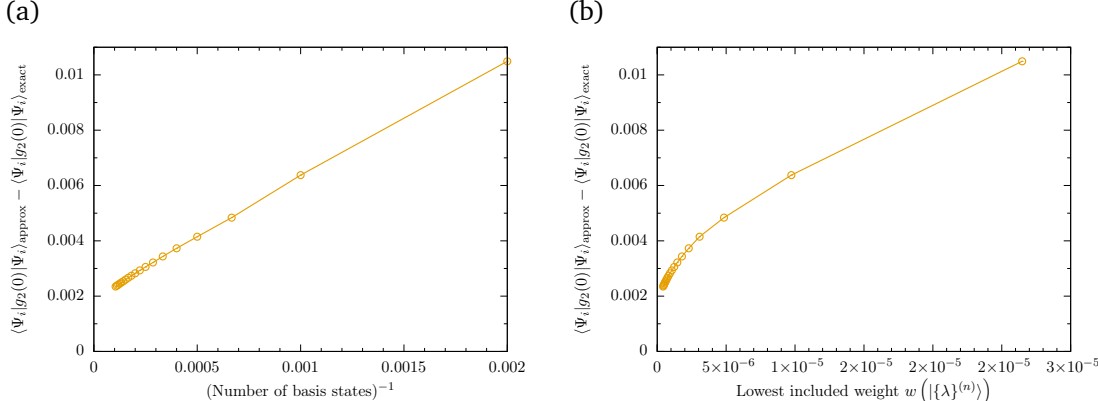

Figure 10: The difference between the expectation value of $g_2(0)$ in the constructed $c_i = 20$ ground state (expressed in terms of $c_f = 10$ eigenstates) and the exact value. Data is presented for $N = 10$ particles, where the exact result is $\langle g_2(0) \rangle_{\text{exact}} = 0.0238263$. (a) Convergence with the number of basis states; (b) convergence with lowest included weight, according to the metric (22).

with the large number of low overlap states (as it was constructed to do), targeting instead the few high overlap states within the energy window of Fig. 5. The truncation scheme is clearly working as desired.

### 3.6.3 Convergence of local expectation values

We have focused thus far on obtaining the energy of the initial state to high precision. One may ask is this convergence criteria is indeed the same as correctly constructing the initial state? In this section we turn our attention towards local properties of the constructed state. In particular, we consider the behavior of expectation values of local operators within both the exact initial state and the approximation initial state. This will allow us to establish that we are correctly reproducing local observables within the state, not only its energy.

To start with our study of local correlations, let us note a trivial point. Particle number $N$ is conserved within the Hamiltonian, which when combined with translational invariance ensures that the expectation value of the local density within all eigenstates (and the approximate initial state) satisfies $\langle \{\lambda\}_N | \Psi^{\dagger}(x)\Psi(x) | \{\lambda\}_N \rangle = N/R$ by construction. Thus our state of course satisfies this restriction, by construction.

Convergence of the energy implies that local expectation values of the operators appearing within the Hamiltonian should also be converging. We confirm this in Figs. 10 and 11, where we present the difference between the constructed and exact values of expectation values of $g_2(0)$ and $\partial_x \Psi^{\dagger}(0)\partial_x \Psi(0)$, respectively. We see that the former operator, $g_2(0)$, is not quite so well converged as $\partial_x \Psi^{\dagger}(0)\partial_x \Psi(0)$. This makes some sense: we construct the state to ensure the energy is well converged, and in the large $c$ limit of the Lieb-Liniger model it is the kinetic energy that dominates the interaction energy (this is particularly apparent in the $c = \infty$ limit, where the model maps to non-interacting fermions). Nonetheless, we do see that we are correctly capturing expectation values of local operators within the constructed states and, when plotted as function of lowest included weight, the convergence to the exact value seems reasonable.

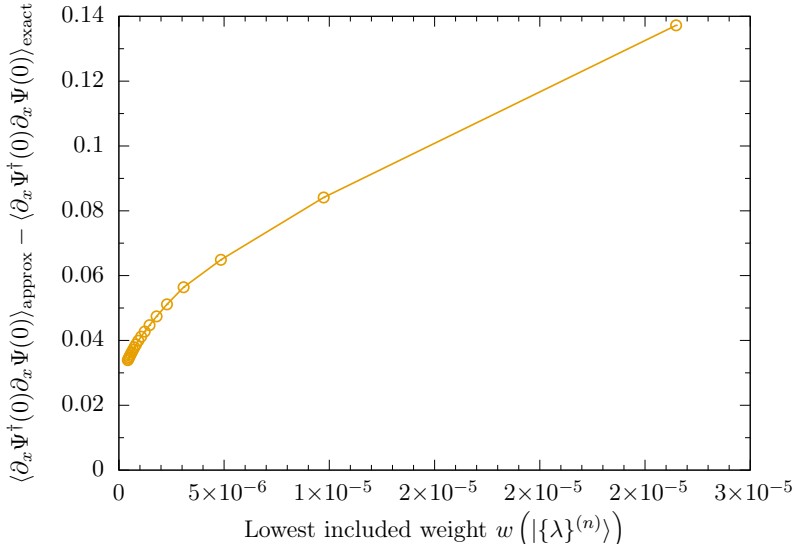

Figure 11: The difference between the expectation value of $\partial_x\Psi^\dagger(0)\partial_x\Psi(0)$ in the constructed $c_i = 20$ ground state (expressed in terms of $c_f = 10$ eigenstates) and its exact value. Data is presented for $N = 10$ particles, where $\langle\partial_x\Psi^\dagger(0)\partial_x\Psi(0)\rangle_{\text{exact}} = 2.22032$.

## 4 Non-equilibrium dynamics from the high overlap states truncation scheme

Having developed the high overlap states truncation scheme, we have so far used it to construct an initial state (motivated by non-equilibrium dynamics) and we have studied the properties of this approximate state. In this section we turn our attention to computing the non-equilibrium dynamics following the $c = c_i \to c_f$ sudden quantum quench. The time evolved state can easily be obtained from Eq. (11), truncated to include $N_{\text{tot}}$ terms via the high overlap states truncation scheme, as in Eq. (14). We use such a representation to first examine the time evolved wave function via the return amplitude and the fidelity, before turning our attention to the time evolution of local observables.

### 4.1 The return amplitude and the fidelity

To begin, we consider a particularly simple quantity to evaluate: the return amplitude

$$\langle\Psi_i|\Psi_i(t)\rangle \approx \sum_{n=0}^{N_{\text{tot}}} e^{-iE(\{\lambda\}^{(n)})t}\left|\langle\{\lambda\}^{(n)}|\Psi_i\rangle\right|^2. \tag{23}$$

This return amplitude has received significant attention in the context of quantum quenches, where it was realized that

$$f(t) = -\lim_{R\to\infty}\frac{1}{R}\log\langle\Psi_i|\Psi_i(t)\rangle, \tag{24}$$

can display non-analytic behavior, related to dynamical quantum phase transitions (see, e.g., [127] for a review and [128] for an example experiment). The absolute value squared of the return amplitude,

$$\mathcal{F}(t) = |\langle\Psi_i|\Psi_i(t)\rangle|^2, \tag{25}$$

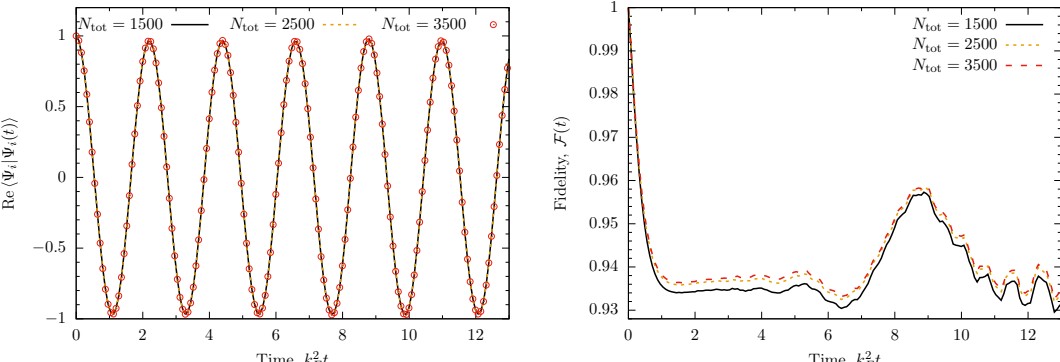

Figure 12: The real part of the return amplitude (23) (left) and the fidelity (25) (right) following the quench $c_i = 20 \to c_f = 10$ in the Lieb-Liniger model, starting from the $c_i$ ground state with $N = 10$ particles at unit density, $N/R = 1$. ($k_F = \pi(N-1)/R$ is the Fermi wave vector in the $c = \infty$ limit.) In both cases, we show the time evolution for three sizes of the truncated Hamiltonian (a matrix of size $N_{\text{tot}} \times N_{\text{tot}}$). Both these quantities rapidly converge with the number of states, see the convergence of the initial state energy for comparison in Fig. 8.

is known as the fidelity.

Here the return amplitude and the fidelity will serve as useful test-beds for understanding the effect truncation of the Hilbert space has on non-equilibrium quantities. This may, in principle, be quite different to the behavior shown in the convergence of the initial state energy studied above. This is because the energies of the eigenstates $|\{\lambda\}^{(n)}\rangle$ entering Eq. (14) are *unbounded*, while each term appearing within the return amplitude (and the fidelity) is bounded. Indeed, we see precisely this difference in Fig. 12, where we show the time evolution of the return amplitude and the fidelity at short times. For small numbers of states in the truncated Hilbert space, $N_{\text{tot}}$, both of these quantities are well converged, unlike the initial state energy for the same number of states, see Fig. 8. This is particularly convenient, as we can achieve excellent convergence of time evolved physical quantities for (very) small number numbers of states.

We note that bench marking convergence of time evolution with the return amplitude, or the fidelity, is also convenient as it involves evaluating only a single sum over the final eigenstates. It can, thus, be evaluated very efficiently even if one requires $N_{\text{tot}}$ large. In the next subsection, we consider time evolution of local observables, which requires evaluating a double sum over the truncated Hilbert space.

## 4.2 Time evolution of local observables

Having examined the return amplitude, which depends solely on the time evolved wave function, we turn our attention to the non-equilibrium behavior of local observables $O$. These are computed by evaluating the double sum over the truncated Hilbert space

$$\langle O(t) \rangle_i \equiv \langle \Psi_i(t)|O|\Psi_i(t)\rangle$$
$$= \sum_{n,m=0}^{N_{\text{tot}}} e^{-it[E(\{\lambda\}^{(n)})-E(\{\lambda\}^{(m)})]}\langle \Psi_i|\{\lambda\}^{(m)}\rangle\langle\{\lambda\}^{(m)}|O|\{\lambda\}^{(n)}\rangle\langle\{\lambda\}^{(n)}|\Psi_i\rangle. \quad (26)$$

Clearly for any observable of interest $O$ we require knowledge of the matrix elements between the Bethe eigenstates, $\langle\{\mu\}^{(m)}|O|\{\lambda\}^{(n)}\rangle$.

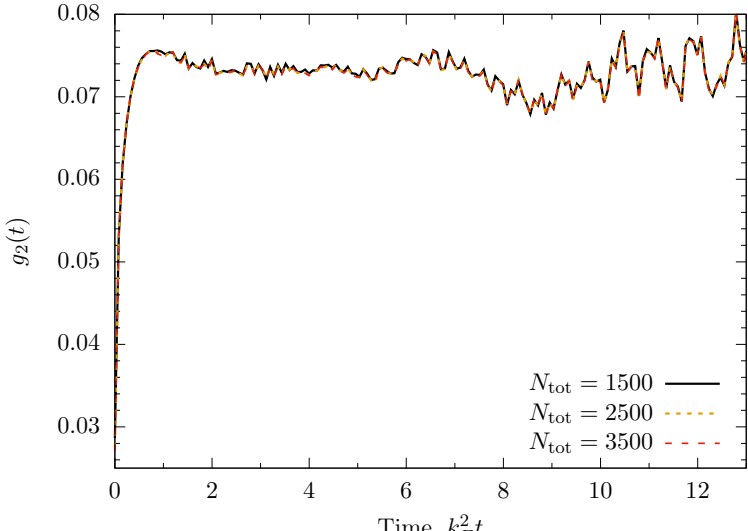

Figure 13: The time evolution of the local observable $g_2 = \left(\Psi^\dagger(0)\right)^2 (\Psi(0))^2$ following a quench $c_i = 20 \rightarrow c_f = 10$ in the Lieb-Liniger model, starting from the ground state at $c_i$ with $N = 10$ particles at unit density, $N/R = 1$. ($k_F = \pi(N-1)/R$ is the Fermi wave vector at $c = \infty$.) Results are shown for a number of different truncated basis sizes, $N_{\text{tot}}$, which illustrates the excellent convergence of $g_2(t)$ for small numbers of states.

Having understood how the truncation of the wave function affects the return amplitude, we check whether the same excellent convergence occurs in the time evolution of local quantities. We focus on the operator $O = g_2(0)$, whose matrix elements are given in Appendix A. Its time evolution, $g_2(t) = \langle\Psi(t)|g_2(0)|\Psi(t)\rangle$, is shown in Fig. 13 for the same quench as before. We observe convergence properties similar to the return amplitude and the fidelity, see Fig. 12, i.e. excellent convergence for small numbers of states within the truncated Hilbert space.

The results of this subsection, taken with those of the previous one, are strongly suggestive that we will be able to efficiently generate the time evolution of observables for relatively large numbers of particles with modest computational resources. We show an example of this in Sec. 4.2.3.

### 4.2.1 Comparison to the coordinate Bethe ansatz

As a check of our results, we turn our attention to results within the literature. In particular, in Ref. [45] a quench of the interaction parameter $c_i = 100 \rightarrow c_f = 3.7660$ was considered via the coordinate Bethe ansatz. This is a large, challenging quench where the interaction parameter changes drastically. The energy density between the initial $c_i$ ground state and the final ground state $c_f$ is significant and presumably many excitations are generated in the quench. This is a challenging scenario for any numerical approach, and it will allow us to assess the precision of our results. (We note that calculations of non-equilibrium time evolution at finite $c_i$ and $c_f$ are very limited in the literature.)

The coordinate Bethe ansatz calculations of Ref. [45] are computational intensive, and scale very poorly with particle number. Data is limited to cases with just small numbers of particles, $N = 5$ in the case at hand [45]. Our algorithm is currently limited to even numbers of particles, to avoid technical issues in dealing with coinciding rapidities that often occur with $N$ odd. Thus we will compare the $N = 5$ results of Ref. [45] to $N = 4$ data obtained within

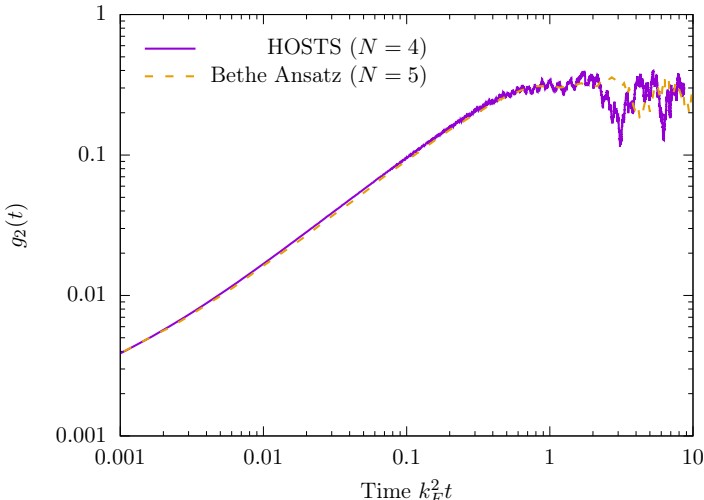

**Figure 14:** The time evolution of the local observable $g_2(0)$ following the $c_i = 100 \rightarrow c_f = 3.7660$ in the Lieb-Liniger model starting from the ground state at $c_i$ with unit density. Exact data (dashed line) computed via the coordinate Bethe ansatz with $N = 5$ particles (from Ref. [45]) is compared to high overlap states truncation scheme (HOSTS) computations with $N = 4$ particles. HOSTS results for higher numbers of particles are discussed in Sec. 5.

our high overlap states truncation scheme (we will discuss $N = 6$ later).

Our comparison to results of the coordinate Bethe ansatz is shown in Fig. 14, where the data from Fig. 4 of [45] was extracted directly from the image. For $N = 4$ particles, our data is generated with $N_{\text{tot}} = 5000$ states via full diagonalization (i.e. we do not need to use the numerical renormalization group). We see excellent agreement up to the finite size revival time. This is promising, as the computational effort within our scheme is rather modest in such a scenario. These results further confirm that the high overlap states truncation scheme is correctly capturing all of the physics within the problem.

### 4.2.2 The long time limit: The diagonal ensemble

Beyond accessing finite time dynamics of observables, we can also access the long-time limit, $t \rightarrow \infty$. Here a number of simplifications occur, as detailed in many works (see, e.g., [15]); for example, with the overlaps at hand, we can compute the diagonal ensemble result for the long-time limit [15]

$$\langle O \rangle_{\text{DE}} = \sum_j \langle \{\lambda\}^{(j)}|O|\{\lambda\}^{(j)}\rangle \left| \langle \{\lambda\}^{(j)}|\Psi_i\rangle \right|^2. \tag{27}$$

This describes the infinite time limit of the time-averaged observables in the large system size limit (see, e.g., the discussion in the appendix of [45])

$$\lim_{T \rightarrow \infty} \frac{1}{T} \int_0^T \mathrm{d}t \langle \Psi_i(t)|O|\Psi_i(t)\rangle \rightarrow \langle O \rangle_{\text{DE}}. \tag{28}$$

If the observable $O$ relaxes to a stationary value in a sufficiently fast manner then that long time limit of the expectation value (without time averaging) will be reproduced

$$\lim_{t \rightarrow \infty} \langle \Psi(t)|O|\Psi(t)\rangle \rightarrow \langle O \rangle_{\text{DE}}. \tag{29}$$

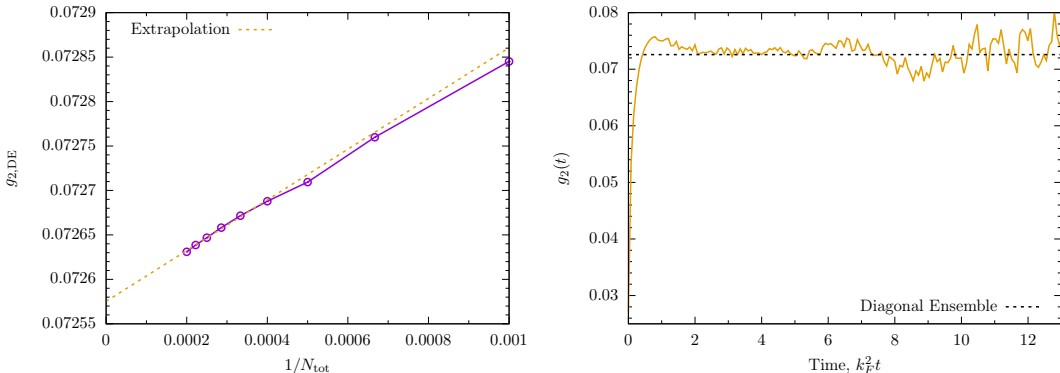

Figure 15: The scaling with $N_{\text{tot}}$ of the diagonal ensemble result (left panel) for $c_i = 20 \rightarrow c_f = 10$ quench with $N = 10$ particles. The dotted line shows a linear extrapolation to $N_{\text{tot}} = \infty$. A comparison of the $N_{\text{tot}} = 3500$ data of Fig. 13 with the extrapolated diagonal ensemble result for the long time limit (right panel).

see, for example, Ref. [129]. As such, the diagonal ensemble can be a computationally convenient way to access the infinite time limit if overlaps are known.

We illustrate that our truncation scheme can evaluate the diagonal ensemble in Figs. 15. We first study the convergence of the diagonal ensemble result as a function of the truncated Hilbert space dimension (left panel), before comparing the extrapolated $N_{\text{tot}} \rightarrow \infty$ result to the real-time dynamics shown in Fig. 13 (right panel). We see well-behaved and rapid convergence of the diagonal ensemble value to its "non-truncated" limit, and that it captures well the values to which local observables relax at intermediate times. We note that fluctuations about the diagonal ensemble value within the real time dynamics are large for the system sizes considered.

We note that expectation values of local operators, long after a quench in an integrable model, are expected to be described by a generalized Gibbs ensemble [6]. In the case of the Lieb-Liniger model, one can run into issues constructing this ensemble because of the asymptotic behavior (in rapidity space) of the steady state root distribution, which leads to diverging expectation values of ultra-local charges. This can, in principle, be remedied by working with a different set of charges (see, e.g., Ref. [48]), but we do not pursue that approach here.

### 4.2.3 An example with larger numbers of particles

So far, we have examined quenches with relatively small numbers of particles, $N = 4, 6, 10$. It is worth emphasizing that even for these numbers of particles, exact calculations via the coordinate Bethe ansatz are computationally expensive, and exact calculations with $N = 10$ corresponding to summing $\propto (N!)^2 \sim 1.3 \times 10^{13}$ terms. To even contemplate exact evaluation for $N = 20$ particles, $\propto 5.9 \times 10^{36}$ terms, seems futile. Instead the high overlap states truncation scheme, in combination with the numerical renormalization group, gives one a handle on such problems.

Here, we consider the $c_i = 20 \rightarrow c_f = 10$ quench for a larger numbers of particles ($N = 20$) as an illustrative example. We leave detailed study, both of larger numbers of particles and different quenches (as well as expanding on results presented above), to future works [130]. Results for the energy convergence and the time evolution of the fidelity, $\mathcal{F}(t)$, are shown in Fig. 16.

(a) 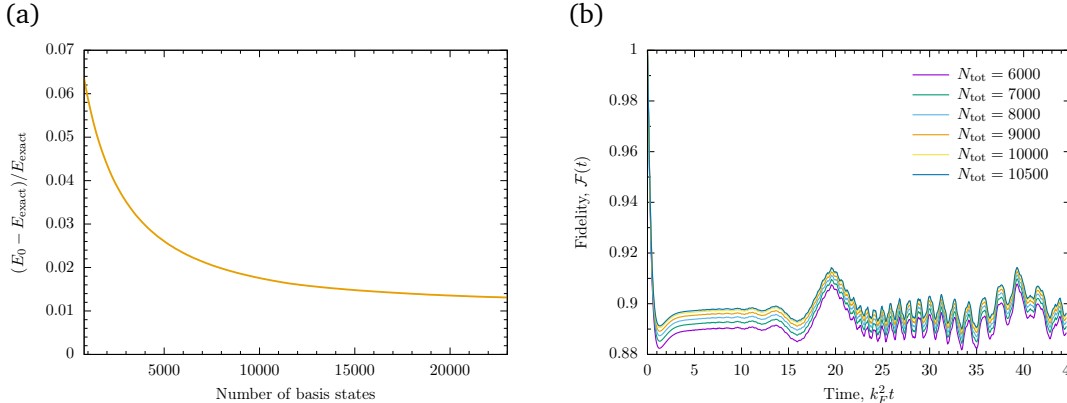 (b)

Figure 16: Example calculation showing (a) the energy convergence of the initial state; (b) the time evolution of the fidelity for $N = 20$ particles, for the quench $c_i = 20 \rightarrow c_f = 10$.

# 5 Introducing the Matrix Element Renormalisation Group

In this section, we consider a perturbing operator $(c_i - c_f)Rg_2(0)$ whose matrix elements, with respect to some of the computational basis states, are large compared to the energy difference between these states and the unperturbed ground state. We call quenches for which the perturbing operator satisfies this property "strongly non-perturbative". In such cases relying solely on the metric (22) discussed in the previous section, which was motivated by leading order perturbation theory, is no longer justified. Higher order terms are expected to be relevant and, as a result, we need to modify the way we select and order states. Not only this, but we need to re-examine the assumptions behind the numerical renormalization group procedure discussed in Sec. 3.3, and modify these accordingly. In the final part of this section we show how the resulting procedure allows us to not only treat quenches where the initial state is a ground state, but also those where it is an excited state.

As matrix elements of the perturbing operator become large, contributions of a given computational basis state to the ground state mediated via other intermediate computational basis states can become relevant. This corresponds to the second order terms in Eq. (21) no longer being negligible compared to the first order terms. However, such contributions are not considered in the standard numerical group procedure as discussed in Sec. 3.3. As a result, these contributions are missed when states are not by chance included in the same step of the renormalization group procedure. We will see that these contributions can play an important role for strongly non-perturbative quenches, so they need to be taken into account.

An illlustration of how naively applying the algorithms developed thus far can lead to inaccurate results for strongly non-perturbative quenches is shown in Fig. 17. The initial steps still correspond to the results obtained from diagonalising the full truncated Hamiltonian, but as the number of iterations increases, the discrepancy becomes larger.

## 5.1 The matrix element renormalization group algorithm for the ground state

To deal with the problem for strongly non-perturbative quenches discussed in the previous section, we develop a reworking of the numerical renormalization group procedure that we refer to as the matrix element renormalization group. The main differences between the matrix element renormalization group and the conventional renormalization group algorithm are:

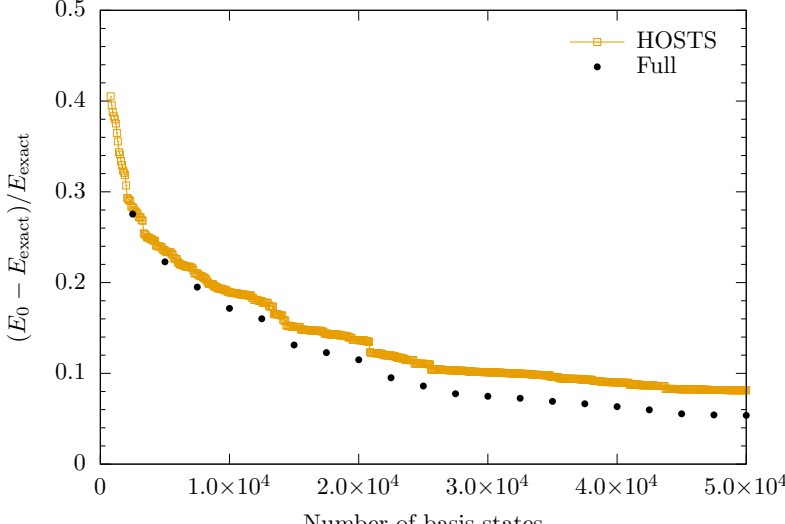

Figure 17: An example calculation showing that for strongly non-perturbative quenches there can be a large discrepancy between the results from full diagonalisation and the NRG-extension of HOSTS. We consider the $c_i = 100 \rightarrow c_f = 3.7660$ quench for $N = 6$ particles, cf. Fig. 14. $N + \Delta N_s = 800$ is fixed within each data set.

1. The approximate eigenstates obtained at each step of the algorithm (from diagonalization of a truncated Hamiltonian) are kept, unlike in the conventional case where one discards $\Delta N_s$ states at each iteration.

2. When introducing new computational basis states, we select which of the previously obtained approximate eigenstates to include in the Hamiltonian using a weighing function based on the quadratic terms in the perturbation-series expansion of the wave function (21) instead of including the approximate eigenstates with the lowest energies.

The matrix element renormalization group takes seriously the idea that matrix elements of the perturbing operator, rather than energies, are the important quantity when operators are not strongly renormalization group relevant. The central idea is that computational basis states $|\{\lambda\}^{(j)}\rangle$ included at a given step can mediate strong coupling between the approximate ground state and approximate "excited states" obtained at an earlier iteration. These "approximate states" must then be included in the truncated Hamiltonian at this diagonalization step to ensure an accurate description of the ground state. So, instead of blindly removing the high energy approximate "excited states" at each step of algorithm (as in the conventional numerical renormalization group), we keep all approximate eigenvectors, and at each iteration include the states most important for mediating the coupling between the approximate ground state and the newly added states from the computational basis.

Let $|\Omega\rangle$ be the ground state of the final Hamiltonian, then the steps of the matrix element renormalization group are as follows:

1. Generate the computational basis via preferential state generation from the ground state $|\Omega\rangle$. Order the states in the computational basis according to the metric in Eq. (22), to obtain $\{|\{\lambda\}^{(j)}\rangle\}$.

2. Construct a truncated Hamiltonian from the first $N_s + \Delta N_s$ computational basis states, $\{|\{\lambda\}^{(1)}\rangle, \ldots, |\{\lambda\}^{(N_s + \Delta N_s)}\rangle\}$ and diagonalize this Hamiltonian to obtain the first approximate eigenstates $\{|E_1\rangle, \ldots, |E_{N_s + \Delta N_s}\rangle\}$ with energies $\{E_1, \ldots, E_{N_s + \Delta N_s}\}$. These approxi-

mate eigenstates replace the first $N_s + \Delta N_s$ states in the computational basis, and are ordered such that $E_1 < \cdots < E_{N_s + \Delta N_s}$.

3. Define a new basis of $N_s + \Delta N_s$ eigenstates for the truncated Hamiltonian by adding the next $\Delta N_s$ states from the computational basis $\{|\{\lambda\}^{(j)}\rangle\}$ to the approximate eigenvector with the lowest energy $|E_1\rangle$ as well as the $N_s - 1$ approximate eigenstates $\{|E_i\rangle\}_{i>1}$ whose "second order weight", given by

$$w_2(|E_i\rangle) = \sum_j \frac{\langle E_i | \delta H | \{\lambda\}^{(j)}\rangle \langle \{\lambda\}^{(j)} | \delta H | E_1\rangle}{(E_1 - E_{\{\lambda\}}^{(j)})(E_1 - E_i)} . \tag{30}$$

is largest. Here $\delta H = cR \times g_2(0)$ is the perturbing operator (see Sec. 2.2.1), the sum ranges over all $\Delta N_s$ newly added computational basis states, and $E_{\{\lambda\}}^{(j)} = E(\{\lambda\}^{(j)})$ are the energies of the newly added computational basis states.

4. Construct the truncated Hamiltonian in this new basis and diagonalize it to obtain $N_s + \Delta N_s$ new approximate eigenstates. These newly constructed approximate eigenstates replace the states in the computational basis used to construct the truncated Hamiltonian.

5. Return to the third step.

This process is continued, obtaining new approximate eigenstates after each cycle of steps 3 to 5, until the required convergence of the ground state energy/eigenstate is reached or the computational basis is exhausted.

The matrix element renormalization group has some slight disadvantages when compared to the conventional numerical renormalization group. Firstly, it is more memory intensive: a complete set of approximate eigenstates must be retained in the procedure, while in the conventional routine we only need keep track of $N_s$ such approximate eigenstates.[9] Secondly, the matrix element renormalization group has a higher computational burden since it requires the computation of the second order weight for all approximate eigenvectors at the start of each iteration. However, we have seen that the conventional numerical renormalization group fails to produce accurate results for strongly non-perturbative quenches (see Fig. 17) so these savings in memory and computations compared to the matrix element renormalization group are moot.

There are a couple of alternative, complementary, schemes that could be used to construct the initial state. Firstly, there exist "sweeping" improvements of the conventional numerical renormalization group (see, e.g., their discussion in [60]). If succesful however, this additional would certainly come at a higher computational cost than directly using the matrix element renormalization group. Secondly, one could invoke iterative diagonalization (via, e.g., Lanczos or Davidson) within a given truncated basis. In such a procedure, one would have to check convergence of results with basis size, but one can (in principle) deal with very large bases. How quickly such iterative diagonalization converges, with our matrix being dense, is not clear. We have yet to explore this avenue, but it is an interesting direction for future works.

## 5.2 Results from the matrix element renormalization group for the ground state

With the matrix element renormalization group algorithm in place, we can employ it to tackle problems that are inaccessible to the conventional numerical renormalization group.

---

[9]In practice, since we need at most $N_s$ approximate eigenvectors at any one time, these eigenvectors do not have to be stored in memory.

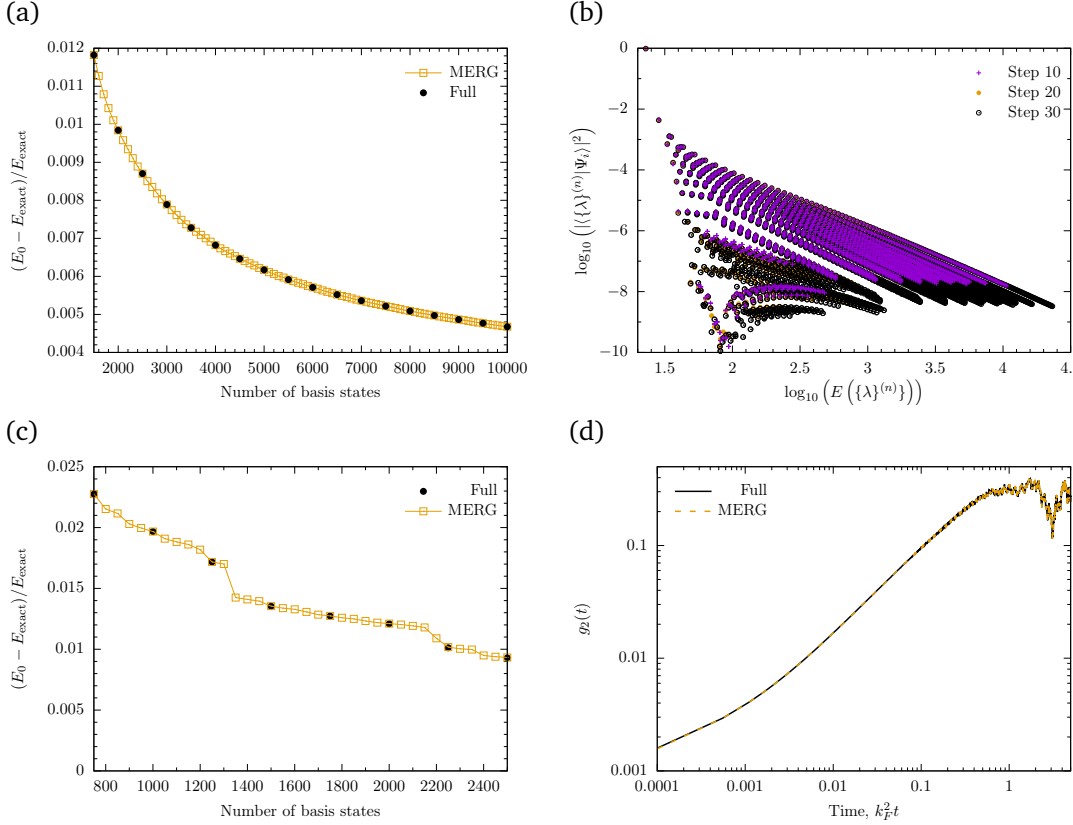

Figure 18: *Top row:* Matrix element renormalization group (MERG) and numerical renormalization group (NRG) for the $c_i = 20 \to c_f = 10$ quench for $N = 10$ particles, computed within the high overlap states truncation scheme. (a) The convergence of the initial state energy as a function of number of basis states; (b) the convergence of the overlaps at different steps of the MERG procedure (cf. Fig. 9). *Bottom row:* MERG and full diagonalization results for the $N = 4$ particle quench $c_i = 100 \to c_f = 3.7660$: (c) the convergence of the energy of the initial state with the number of basis states; (d) the time evolution of $g_2(t)$ with 2500 basis states.

However, in the first case, we check that the matrix element renormalization group correctly reproduces results in cases where the conventional numerical renormalization group approach works. This is a basic sanity check: can we reproduce the initial state and its dynamics in these simpler cases. Our first example is $c_i = 20 \to c_f = 10$ quench studied earlier in this work. We present the convergence of the energy and the overlaps in Figs. 18(a)–(b). In particular, Fig. 18(b) should be compared to Fig. 9 obtained previously. We see excellent agreement between the conventional numerical renormalization group and the matrix element renormalization group in this scenario.

As a second check, we turn our attention to the harder quench considered in the previous section for $N = 4$ particles, $c_i = 100 \to c_f = 3.7660$. Here we check against full diagonalization of the truncated Hamiltonian (as the required number of states for excellent convergence is rather small), as shown in Figs. 18(c)–(d). The matrix element renormalization group gives results in excellent agreement with full diagonalization of the same basis, both in terms of energy of the initial state, Fig. 18(c), and the non-equilibrium dynamics of observables, Fig. 18(d).

With the matrix element renormalization group correctly reproducing both full diagonal-

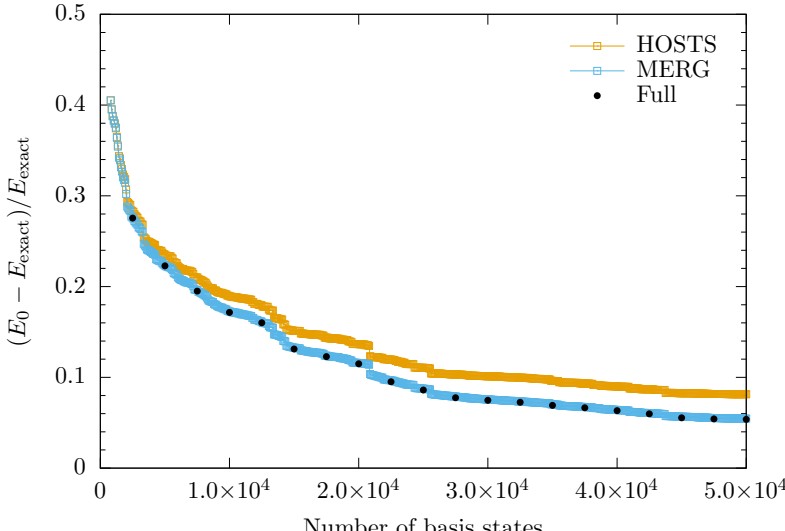

Figure 19: The convergence of the initial state energy $E_0$ for $N = 6$ particle quench $c_i = 100 \rightarrow c_f = 3.7660$ obtained with the matrix element renormalization group (MERG). MERG is performed with $N_s = 720$ and $\Delta N_s = 80$. Conventional numerical renormalization group approaches breakdown in this scenario (for the same $N_s, \Delta N_s$) as shown in Fig. 17. Full diagonalization results, for the same number of basis stats, are shown for comparison.

ization (in small bases) and conventional numerical renormalization group (in large bases) results, we examine the problematic scenario discussed in the previous section. In this strongly non-perturbative quench, the matrix element renormalization group is vital for correctly constructing the initial state. In scenarios where the conventional numerical renormalization group fails to produce results which agree with results obtained by diagonalization of the full truncated Hamiltonian, such as the one illustrated in Fig. 17, the matrix element renormalization group continues to produce results that agree with great accuracy as is shown in Fig. 19.

In Fig. 19 we see a number of features. Firstly, we note that the agreement between the results obtained by full diagonalisation of the truncated Hamiltonian and using the matrix element renormalization group are in excellent agreement. Secondly, regardless of the procedure, the convergence of the initial state energy shows some plateaus and jumps, which implies that the metric (22) is not the perfect one. Understanding how to construct the most convergent metric for a given problem is an outstanding challenge, which requires further investigations. Thirdly, we see that for $N = 6$ particles the problem is very challenging: By including 50,000 states, we still only achieve initial state energies correct to within $\sim 5.5\%$ ($\sim 10\%$ w.r.t. the Fermi energy). Whilst a better ordering metric might improve this, it still seems likely that strongly nonperturbative quenches will present a significant numerical challenge. This is further supported by Fig. 20, where we show the time evolution of $g_2(t)$ as compared to results from the coordinate Bethe ansatz discussed previously. We see that even a truncated wave function (14) with 50,000 states included does not accurately realize the short time dynamics of observables. At longer times, once the steady state plateau is approached (and high energy modes have dephased, effectively averaging to zero), the truncated wave function does describe $g_2(t)$ well.

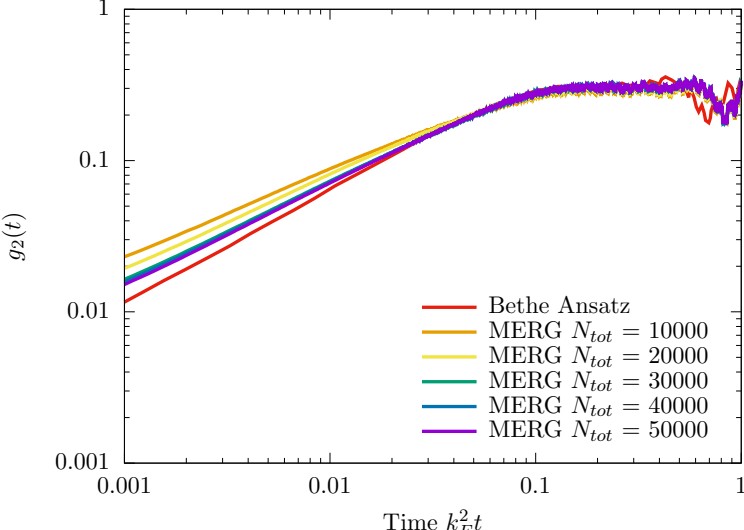

Figure 20: The time evolution of the local observable $g_2(0)$ following the $c_i = 100 \rightarrow c_f = 3.7660$ in the Lieb-Liniger model starting from the ground state at $c_i$. Exact data (dashed line) computed via the coordinate Bethe ansatz with $N = 5$ particles (from Ref. [45]) is compared to matrix element renormalization group (MERG) calculations with $N = 6$ particles.

## 5.3 The matrix element renormalization group algorithm for excited states

In contrast to conventional renormalization group techniques, which at most target the first few excited states in addition to the ground state, the matrix element renomalization group can also be used to target more highly excited states. Furthermore, the algorithm can construct these states without the need to construct all the states of lower energy making the procedure a more efficient tool to target excited states than the methods discussed thus far. In order to do so we have to make some changes to the algorithm described in Sec. 5.1.

To understand why we need to change the algorithm in Sec. 5.1 in order to consider excited states, let us consider what happens if we replace the ground state with an excited state in the algorithm. This state will henceforth be referred to as the seed state. First of all, the preferential state generation routine leads to a different computational basis, as we now consider $|\{\lambda\}^{(0)}\rangle$ in Eq. (22) to be an excited state. In particular, the ground state may not have a high weight according to this metric, so it may not even be included in the computational basis obtained via preferential state generation. Second of all, the algorithm retains the lowest energy approximate eigenstate $|E_1\rangle$ at every step, and selects the approximate eigenstates most relevant to this approximate eigenstate based on the second order weight. This means that the selection rules are still set to promote the convergence of the lowest energy eigenstate, rather than an excited state. Finally, it is generally unclear which eigenstate of the perturbed Hamiltonian corresponds to which of the approximate eigenstates obtained by the algorithm, as energies of different excited states can be very similar and even degenerate.

Before we discuss how we resolve these issues, note that every step of the algorithm represents a mapping between the states used to construct the truncated Hamiltonian and the approximate eigenstates obtained by diagonalization. To identify which of the approximate eigenstates a given state used in the basis for the truncated Hamiltonian is mapped to, we compute the overlaps between this state and all newly obtained approximate eigenstates. The approximate eigenstate with the largest overlap is then said to be its image provided that the

RG-step is "small enough". This allows us to track the approximate eigenstate derived from the seed state throughout the procedure.

The main assumption behind our method of tracking the seed state is that the number of states added at every step of the routine is small enough, so that no single iteration wildly changes the (image of) the seed state. What step size is small enough for this assumption to hold depends on the quench and seed state under consideration.[10] However, there are some general methods by which one can check if an appropriate step size has been chosen. Firstly, one can consider the overlaps computed at each iteration of the routine and verify that there is only one state with a significant overlap. Secondly, one can rerun the routine with a smaller step size and check that it produces the same results. With the preferential scanning routine in place, by which the most significantly states are identified and included first, the start of the routine is where the changes are most drastic and therefore the procedure is most likely to break down there. As a result, the checks proposed here need not be time-consuming.

Now that we have established how we can track the seed state, we note that we can replace the lowest energy approximate eigenstate with the image of the seed state in the second order metric used in Eq. (31). This change, together with the replacement of the ground state with an arbitrary seed state in the preferential scanning routine results in a routine designed to optimize the convergence of the approximate eigenstate associated to the seed state. The resulting algorithm can be summarized as follows.

Let $|\Omega\rangle$ be some eigenstate of the final Hamiltonian, which in this case can be an excited state, then the steps of the matrix element renormalization group are as follows:

1. Generate the computational basis via preferential state generation from the seed state $|\Omega\rangle$. Order the states in the computational basis according to the metric in Eq. (22), to obtain $\{|\{\lambda\}^{(j)}\rangle\}$.

2. Construct a truncated Hamiltonian from the first $N_s + \Delta N_s$ computational basis states, $\left\{|\{\lambda\}^{(1)}\rangle, \ldots, |\{\lambda\}^{(N_s+\Delta N_s)}\rangle\right\}$ and diagonalize this Hamiltonian to obtain the first approximate eigenstates $\left\{|1\rangle, \ldots, |N_s + \Delta N_s\rangle\right\}$ with energies $\{E_1, \ldots, E_{N_s+\Delta N_s}\}$.

3. Compute the overlaps between $|\{\lambda\}^{(0)}\rangle$ and the newly acquired approximate eigenvectors $\{|1\rangle, \ldots, |N_s + \Delta N_s\rangle\}$. Then relabel the approximate eigenstates such that $|1\rangle$ refers to the approximate eigenstate with the largest overlap.

4. Take the next $\Delta N_s$ computational basis states $|\{\lambda\}^{(j)}\rangle$ and compute the "second order weight" for each of the approximate eigenstates $|i\rangle$ with $i > 1$ obtained in previous steps:

$$w_2(|i\rangle) = \sum_j \frac{\langle i|\delta H|\{\lambda\}^{(j)}\rangle\langle\{\lambda\}^{(j)}|\delta H|1\rangle}{(E_1 - E_{\{\lambda\}}^{(j)})(E_1 - E_i)}. \tag{31}$$

   Here $\delta H = cR \times g_2(0)$ is the perturbing operator (see Sec. 2.2.1), the sum ranges over all the newly added computational basis states, and $E_{\{\lambda\}}^{(j)} = E(\{\lambda\}^{(j)})$ are the energies of the newly added computational basis states.

5. Form a truncated basis consisting of $|1\rangle$, the $N_s - 1$ states in $\{|i\rangle\}_{i>1}$ with the largest $w_2$-weight, and the $\Delta N_s$ computational basis states introduced in step 4, construct the truncated Hamiltonian in this basis and diagonalize it to obtain $N_s + \Delta N_s$ new approximate eigenstates $\{|1'\rangle, \ldots, |(N_s + \Delta N_s)'\rangle\}$.

---

[10]It may happen that no step size is small enough when we consider a very strong quench and/or a highly excited state.

6. Compute the overlaps between $|1\rangle$ and the newly acquired approximate eigenvectors, and replace $|1\rangle$ by the approximate eigenvector with the largest overlap. Replace the remaining approximate eigenvectors used to form the truncated basis in step 5 with the remaining newly obtained approximate eigenvectors.

7. Return to the fourth step.

This process is continued, obtaining new approximate eigenstates after each cycle of steps 4 to 6, until the required convergence of the eigenstate is reached or the computational basis is exhausted.

This version of the matrix element renormalization group is not more memory intensive than the routine presented for constructing ground states and it is only slightly more computationally intensive. The additonal computational cost comes from computing the overlaps at each iteration.

## 5.4 Results from the matrix element renormalization group for excited states

As mentioned in Sec. 5.3, one of the subtleties that arises when considering the matrix element renormalization group for excited states is that, even though we know that we construct an approximate eigenstate of the perturbed Hamiltonian, we do not necessarily know a priori what eigenstate this will correspond to. For the interaction quench considered here, the most natural eigenstate of $H(c_i)$ to end up with when starting from an eigenstate of $H(c_f)$ is the eigenstate with the same quantum numbers. In this section we show some preliminary results to verify this claim, although we do note that to assert with more certainty that this claim is true, more properties of the eigenstates other than the energies would have to be considered. This is left to future works.

Consider again the quench from $c_i = 20$ to $c_f = 10$. In the following we present the results obtained from running the algorithm three times with three different seed states, whose doubled quantum numbers are given by

$$\text{State A: } \{-9, -7, -5, -3, -1, 1, 3, 5, 7, 9\}, \tag{32}$$

$$\text{State B: } \{-11, -7, -5, -3, -1, 1, 3, 5, 7, 11\}, \tag{33}$$

$$\text{State C: } \{-17, -13, -9, -5, -1, 1, 5, 9, 13, 17\}. \tag{34}$$

The results for the energy convergence of the approximate eigenstates corresponding to these seed states obtained from the matrix element renormalization group are shown in Fig. 21. In order to keep track of the convergence, we again consider the percentual error of the energy only this time with respect to a different target energy of the each of the runs. The target energy $E_{exact}$ is taken to be the energy of the eigenstate of $H(c_i)$ with quantum numbers identical to those of the seed state under consideration. We note that even though we only consider data for the energy convergence here, we have still computed the expansion of the approximate eigenstates in terms of the eigenstates of the intiial basis, so could still compute the time evolution of operators if we please to do so.

The rate of convergence of the run seeded by the lowest excited state, state B, is comparable to the convergence when considering the ground state, state A. On the other hand, when considering a run seeded by a highly excited state, state C, the convergence shows characteristics reminding us of the strongly non-perturbative quenches considered in section Sec. 5.2. Also, in order to keep track of the right approximate eigenstate throughout the procedure, we had to significantly alter the parameters characterizing the size of the renormalization group steps to $N_s = 75$ and $\Delta N_s = 25$.

The fact that considering higher energy states requires a decrease in step size is what currently limits how high the energy of the seed states may be. To overcome this limitation,

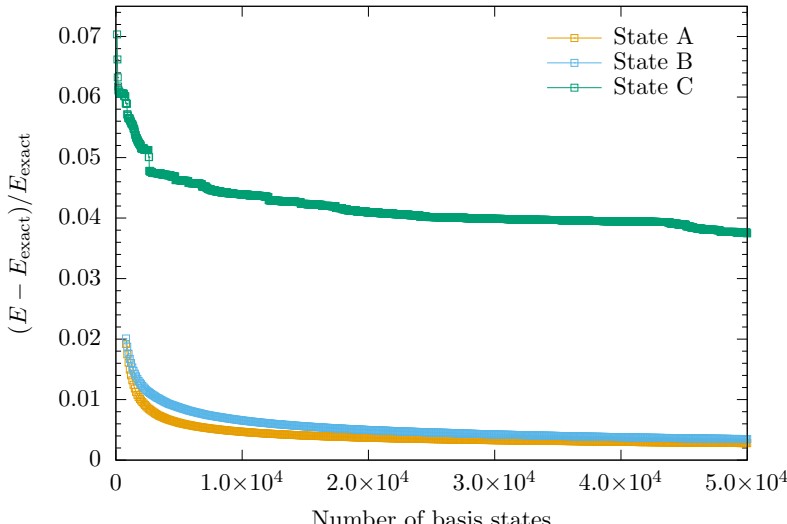

Figure 21: The convergence of the states A, B, and C, for the $N = 10$ particle quench $c_i = 20 \to c_f = 10$ obtained using the matrix element renormalization group (MERG). For state A, and B, MERG is performed with $N_s = 700$ and $\Delta N_s = 100$, whereas for state C, MERG is performed with $N_s = 75$ and $\Delta N_s = 25$.

we would have to reorder the computational basis so that single steps of the procedure do not change the targeted approximate eigenstate as violently. Nevertheless, even without such alterations, our current algorithm goes beyond what one could target using the conventional renormalization group techniques we discussed at the start of this paper, because in that case one would have to construct all lower energy eigenstates.

## 6 Conclusions

Even in the presence of integrability, the computation of non-equilibrium dynamics following a quantum quench remains a great challenge for theory. Well-controlled numerical approaches are vital for accessing the physics away from analytically tractable limits, including for the cases of finite-time dynamics of observables. Here we have presented a proof-of-principle investigation of finite-$c$ to finite-$c$ quenches in the Lieb-Liniger model using a high overlap states truncation scheme, in combination with full diagonalization, the numerical renormalization group, and a new matrix element renormalization group algorithm. We have worked with interacting computational basis states, which intrinsically have built-in strong correlations, and we have systematically constructed initial states in terms of high overlap states, for quenches starting from ground states as well as excited states. Using these, we have computed both real-time dynamics and the long-time limit of physical observables following a quench.

In our development of a high overlap states truncation scheme, and the matrix element renormalization group, we have highlighted the important role played by the ordering of the computational basis. Applying the conventional metric, energy of the computational basis states, we observe poor convergence of properties of the initial states. This poor convergence means applying conventional "truncated spectrum methods" (in their naive form) requires the use of unfeasibly many computational basis states. By modifying the metric, to a "matrix element" focused one that takes into account the structure of the operator coupled to the quench parameter, we achieve orders-of-magnitude improvement in the convergence of properties of

the initial state with truncated Hilbert space size. This was studied in detail in Sec. 3. Along the way, we were able to develop a routine that preferentially generates the states with high overlap following a quench, and this enabled efficient convergence of the initial state energy to sub-percent precision.

This improved convergence opened the door to computing non-trivial non-equilibrium dynamics for numbers of particles far beyond the reach of brute force computations. This was discussed in Sec. 4. Convergence of real-time non-equilibrium dynamics of local observables with the number of computational basis states was surprisingly fast: for $N = 10$ particles $c_i = 20 \rightarrow c_f = 10$ quench, well-converged results for time evolution of $g_2(0)$ are obtained with just thousands of states (some of which are of very high in energy). The long-time limit was also shown to be efficiently accessed via the diagonal ensemble, with results agreeing with the intermediate time dynamics, as expected.

In the case of strongly non-perturbative quenches, we found that conventional numerical renormalization group improvements have to be significantly modified to achieve accurate results. This modification, the so-called matrix element renormalization group, takes seriously that the properties of the perturbing operator should govern the whole procedure. We found this modification to be necessary in the "large quench" studied previously in the literature [45], $c_i = 100 \rightarrow c_f = 3.7660$, when considering more than four particles. Our results were compared to the coordinate Bethe ansatz results of Zill *et al.* [45], and were found to be in excellent agreement. We note, however, that such strongly non-perturbative quenches remain challenging problems, with the quench projecting the initial state on to many states with sizable overlaps. This makes it tough to tackle even relatively small numbers of particles, even with our computationally efficient approach. This seems like an insurmountable problem without introducing additional approximations, beyond the scope of this work, or alternative Hilbert space ordering metrics.

Finally, we considered the construction of excited states of the perturbed Hamiltonian. We wrote down a variant of the matrix element renormalization group algorithm able to directly construct excited states of a perturbed Hamiltonian in terms of the eigenbasis of another Hamiltonian without having to construct all lower energy eigenstates. This allows one to consider more highly excited states than one normally could using the truncated spectrum approach and focusses the computational resources on this particular eigenstate rather than it being a less well-converged side-product of trying to converge the approximate eigenstate representing the ground state of the perturbed Hamiltonian.

The presented high overlap states truncation scheme, combined with full diagonalization and renormalization group improvements, can be applied to many other models and scenarios. Perhaps the most interesting is to consider the case with integrability-breaking where, provided matrix elements of the integrability-breaking terms are known, one can directly apply the same approach. This enables, for example, non-perturbative studies of prethermalization (see, e.g., Refs. [8,28–31]) in continuum quantum gases. Other interesting directions include extensions to other integrable continuum models, such as two-component Bose and Fermi gases or the sine-Gordon regime away from the ultra-relevant perturbation limit [67]. Finally, we would like to point out that the method developed in this paper provides, in principle, all the ingredients necessary to compute for example the time evolution of the entanglement entropy. In order to come to a tractable computation one can convert the overlaps coming from the NRG-routines to a root distribution and then use the quasi-particle picture formulas for the entanglement entropy, see e.g. [131,132]. However, in order to ascertain the accuracy of results obtained in this way, a careful quantitative study of finite-size effects is required in order to determine if we can accurately match results in the thermodynamic and scaling limits. We leave addressing this challenge to future work.

Extending these methods to lattice models should also be possible, using strongly corre-

lated integrable eigenstates. Such an algorithm may complement existing ones: being able to tackle longer times, but smaller systems, than the time-dependent density matrix renormalization group, but larger system sizes than exact diagonalization. It may also be interesting to implement the ideas behind the matrix element renormalization group to lattice and impurity models, invoking a Wilsonian numerical renormalization group-like picture with strongly correlated basis states. These points remain for future works.

The approach implemented within this work for simulating continuum one-dimensional models provides an alternative, complementary approach to continuum matrix product state methods [133, 134]. Utilizing the solvability of a proximate integrable point, time evolution is easy within our approach and can be performed to long times with high precision. This opens the door to novel, non-perturbative studies of non-equilibrium dynamics in models of relevance to cold atomic gases.

# Acknowledgments

We are grateful to Stijn de Baerdemacker, Andrew James, Robert Konik, Gábor Takács and Matthew Walters for useful and inspiring conversations over the past two years while we explored these ideas. Thanks also go to Matthew Davis and Jan Zill for useful correspondence regarding previous coordinate Bethe ansatz calculations of non-equilibrium dynamics, Ref. [45], presented in Fig. 14.

# Funding Information

This work received funding from the European Union's Horizon 2020 research and innovation programme under grant agreement No 745944 (N.J.R) and the European Research Council under ERC Advanced grant No 743032 DYNAMINT (all authors).

# A   Matrix elements

In the main text, we have made use of many known expressions for matrix elements of operators. In this appendix, we provide a summary of these results taken from Refs. [74, 75, 78, 79].

## A.1   Matrix elements of $\Psi^\dagger(0)\Psi(0)$: determinant representation

Expectation values of the density operator $\Psi^\dagger(0)\Psi(0)$ are fixed by the U(1) number conservation and translational invariance to read

$$\frac{\langle\{\mu\}_N|\Psi^\dagger(0)\Psi(0)|\{\mu\}_N\rangle}{\langle\{\mu\}_N|\{\mu\}_N\rangle} = \frac{N}{R}. \tag{35}$$

That is, the expectation value is simply the average density.

Off-diagonal matrix elements can be expressed in terms of a single determinant, as can easily be obtained from [75]. These read:

$$\langle\{\mu\}_N|\Psi^\dagger(0)\Psi(0)|\{\lambda\}_N\rangle = i\mathcal{J}_1(\{\mu\}_N,\{\lambda\}_N)\prod_{j=1}^N\left(V_j^+ - V_j^-\right)\prod_{j,k=1}^N\left(\frac{\lambda_j - \lambda_k + ic}{\mu_j - \lambda_k}\right)$$

$$\times \frac{\det\left(\delta_{jk} + U_{jk}(\lambda_p)\right)}{V_p^+ - V_p^-}, \tag{36}$$

where, explicitly, $\{\mu\}_N \neq \{\lambda\}_N$. In the above, we use the following notations for functions and matrices:

$$\mathcal{J}_1 = P(\{\lambda\}_N) - P(\{\mu\}_N), \tag{37}$$

$$V_j^{\pm} = \prod_{m=1}^{N} \frac{\mu_m - \lambda_j \pm \mathrm{i}c}{\lambda_m - \lambda_j \pm \mathrm{i}c}, \tag{38}$$

$$U_{jk}(\lambda_p) = \frac{\mathrm{i}}{V_j^+ - V_j^-} \frac{\prod_{m=1}^{N}(\mu_m - \lambda_j)}{\prod_{m \neq j=1}^{N}(\lambda_m - \lambda_j)} \big[ K(\lambda_j, \lambda_k) - K(\lambda_p, \lambda_k) \big], \tag{39}$$

and $K(\lambda_j, \lambda_l)$ is given in Eq. (7). Notice that $\mathcal{J}_1$ implies that the matrix element element of the density operator between non-identical states within the same momentum sector vanish.

## A.2  Off-diagonal matrix elements of $g_2(0)$: determinant representation

An efficient, single determinant representation for the off-diagonal matrix elements of $g_2(0)$ is provided by Piroli and Calabrese [79]:

$$\langle \{\mu\}_N | g_2(0) | \{\lambda\}_N \rangle = \frac{(-1)^N}{6c} \mathcal{J}_2(\{\mu\}_N, \{\lambda\}_N) \prod_{j=1}^{N} \left( V_j^+ - V_j^- \right) \prod_{j,k=1}^{N} \left( \frac{\lambda_j - \lambda_k + \mathrm{i}c}{\lambda_j - \mu_k} \right)$$

$$\times \frac{\det_N(\delta_{jk} + U_{jk}(\lambda_p, \lambda_s))}{(V_p^+ - V_p^-)(V_s^+ - V_s^-)}. \tag{40}$$

Here explicitly: (i) the sets of rapidities do not coincide ($\{\mu\}_N \neq \{\lambda\}_N$); (ii) no individual elements of the sets coincide ($\mu_j \neq \lambda_k \; \forall j, k$). In Eq. (40) the following functions and matrices are required:

$$\mathcal{J}_2 = \big[ P(\{\lambda\}_N) - P(\{\mu\}_N) \big]^4 + 3 \big[ E(\{\lambda\}_N) - E(\{\mu\}_N) \big]^2$$

$$- 4 \big[ P(\{\lambda\}_N) - P(\{\mu\}_N) \big] \big[ Q_3(\{\lambda\}_N) - Q_3(\{\mu\}_N) \big], \tag{41}$$

$$U_{jl}(\lambda_p, \lambda_s) = \frac{\mathrm{i}}{V_j^+ - V_j^-} \frac{\prod_{m=1}^{N}(\mu_m - \lambda_j)}{\prod_{\substack{m=1, \\ m \neq j}}^{N}(\lambda_m - \lambda_j)} \big[ K(\lambda_j, \lambda_l) - K(\lambda_p, \lambda_l) K(\lambda_s, \lambda_j) \big]. \tag{42}$$

Furthermore $V_j^{\pm}$ is given in Eq. (38), $K(\lambda_j, \lambda_l)$ is defined in Eq. (7), $Q_3(\{\lambda\})$ is given by Eq. (4), and $\lambda_p, \lambda_s$ are *arbitrary* complex numbers.

Within the main text, we consider states within the same momentum sector, where

$$\mathcal{J}_2(\{\mu\}_N, \{\lambda\}_N) \Big|_{P(\{\mu\}_N) = P(\{\lambda\}_N)} = 3 \big[ E(\{\lambda\}_N) - E(\{\mu\}_N) \big]^2. \tag{43}$$

## A.3  Matrix elements of $g_K(0)$

We now recount known results for matrix elements of the operator

$$g_K(0) = \left( \Psi^\dagger(0) \right)^K \left( \Psi(0) \right)^K, \tag{44}$$

as derived by Pozsgay [78]. We have implemented these expression both for the diagonal elements of $g_2(0)$.

### A.3.1 Diagonal elements

The diagonal elements of $g_K(0)$ are given by:

$$\frac{\langle\{\lambda\}_N|g_K(0)|\{\lambda\}_N\rangle}{\langle\{\lambda\}_N|\{\lambda\}_N\rangle} = (K!)^2 \sum_{\substack{\{\lambda^+\}\cup\{\lambda^-\}\\ |\{\lambda^+\}|=K}} \left[\prod_{j>l}\frac{\lambda_j^+ - \lambda_l^+}{(\lambda_j^+ - \lambda_l^+)^2 + c^2}\right] \times \frac{\det\mathcal{M}}{\det\mathcal{N}}, \tag{45}$$

where $\mathcal{M}$ is an $N \times N$ matrix with elements

$$\mathcal{M}_{jl} = \begin{cases} (\lambda_j)^{l-1}, & \text{for } l = 1,\dots,K, \\ \mathcal{N}_{jl} & \text{for } l = K+1,\dots,N. \end{cases}, \tag{46}$$

and $\mathcal{N}$ is the Gaudin matrix, see Eq. (6). Here it should be understood that the rapidities are ordered as $\{\lambda\} = \{\{\lambda^+\}, \{\lambda^-\}\}$. Whilst this is a sum of determinants, so not as computationally efficient to evaluate as the previous single determinant representation, it is still relatively easy to compute numerically.

### A.3.2 Off-diagonal elements

The off-diagonal matrix elements for $g_K(0)$ read:

$$\langle\{\lambda\}_N|g_K(0)|\{\mu\}_N\rangle = c^K(K!)^2 \sum_{\substack{\{\lambda^+\}\cup\{\lambda^-\}\\ |\{\lambda^+\}|=K}} \left(\prod_{o,\ell}\frac{\lambda_o^- - \lambda_\ell^+ + \mathrm{i}c}{\lambda_o^- - \lambda_\ell^+}\right)$$

$$\times \frac{\prod_{i,j}(\lambda_i - \lambda_j^- + \mathrm{i}c)}{\prod_{m<n}(\mu_m - \mu_n)\prod_{r<s}(\lambda_r^- - \lambda_s^-)} \det\mathcal{W}, \tag{47}$$

where $\mathcal{W}$ is an $N \times N$ matrix with elements

$$\mathcal{W}_{j,l} = (\mu_j)^{l-1}, \qquad \text{for } l = 1,\dots,K, \tag{48}$$

$$\mathcal{W}_{j,K+l} = \frac{\mathrm{i}c}{(\mu_j - \lambda_l^-)(\mu_j - \lambda_l^- + \mathrm{i}c)}$$

$$+ \frac{\mathrm{i}c}{(\lambda_l^- - \mu_j)(\lambda_l^- - \mu_j + \mathrm{i}c)}\prod_{o=1}^N\frac{(\lambda_l^- - \mu_o + \mathrm{i}c)(\lambda_l^- - \lambda_0 - \mathrm{i}c)}{(\lambda_l^- - \mu_o - \mathrm{i}c)(\lambda_l^- - \lambda_o + \mathrm{i}c)}, \tag{49}$$

and the Bethe states are normalized as in Eq. (5).

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
