# Peer review of "On computing non-equilibrium dynamics following a quench"

_SciPost Physics, doi:SciPost Phys. 11, 104 (2021)_

## Round 2 · Referee Report · Anonymous (Referee 1) · 2020-12-23

Strengths
Development of a new numerical method with potentially many applications
Weaknesses
I do not like to list weaknesses
Report
In this paper, the authors introduce a new numerical approach to study the quench dynamics of integrable systems (with the possibility to extend to non integrable ones). The method is non-perturbative and it is inspired by TCSA in field theory, but it goes beyond it in many respects: 1) it preferentially generates the states with high overlaps for a quantum quench 2) it uses the basis of an interacting model 3) it encapsulates the most relevant states by means of a “high overlap states truncation scheme”. The method explained in a very detailed manner with a large number of convincing tests for a prototypical quench within the Lieb-Liniger model. It is surely not very useful to further summarise the results of a 50 pages long paper here, thus I move directly to the comments
The paper is well written and original. The ideas are extremely interesting and have a wide range of applications. Some results are presented for a specific interaction quench in the Lieb-Liniger model, but this is only one of the many possible applications. Similar ideas can be used beyond the Bose gas, for example also in the presence of string states and for non-integrable Hamiltonians, although there are some technical difficulties to overcome before addressing also these interesting problems. However, l I believe that the method is of large interest for the community even if it would only apply to the Bose gas, as proved here.
In conclusion, I think this paper is very good, and should be published in SciPost. Anyhow, I have a minor proposal that the authors could consider to implement or at least comment before publication (I do not need to see the paper again, I am sure that the authors reaction will be appropriate).
From the numerically calculated overlaps, the authors can easily construct a Bethe representative state (in the quench action language) and from this use the quasiparticle picture to reconstruct the time evolution of the entanglement entropy. I think that the addition of this result (here or in another paper if it takes too long) will nicely complement the presented predictions for local observables. Furthermore, they can also trivially determine the diagonal entropy and explore the relation between diagonal and thermodynamic entropy in the case the post-quench states are not parity invariant.
Requested changes
See above
Author: Albertus de Klerk on 2021-02-16 [id 1241]
(in reply to Report 1 on 2020-12-23)
We thank the referee for their careful reading, and agree with the summary of the paper. Furthermore, we thank the referee for their support for our work and their interesting suggestions.
In their report the referee makes the interesting suggestion to consider including the time evolution of the entanglement entropy as well as the diagonal entropy, and the relation between the diagonal and the thermodynamic entropy in the situation where post-quench states are not parity-invariant. We will address these suggestions in turn below.
Although the referee is right that in principle we have all the ingredients necessary to compute the time evolution of the entanglement entropy, the computation itself presents a significant computational challenge. This is due to the fact that the position representation of the Bethe states consists of a sum over N! terms, where N is the number of particles. Therefore the computation of the reduced density matrix of a subsystem at a given moment in time involves a summation over N2tot∗(N!)2 terms, where Ntot is the size of the truncated basis used throughout the paper. Even for a system consisting of only 10 particles, and a basis size of 1000 states, this would require the summation over 1.3*10^19 terms for the computation of the reduced density matrix at a given moment in time. Furthermore, it would have to be verified that the truncation scheme we devised in our work is also appropriate for quantities such as the entanglement entropy.
Given these challenges, we think the resolution of these issues and a detailed study of entropic considerations would be more appropriate for future works. We have incorporated a comment regarding the suggestion of the referee in the conclusions section of the paper that reads as follows: "Finally, we would like to point out that the method developed in this paper provides, in principle, all the ingredients necessary to compute for example the time evolution of the entanglement entropy. However, the computation of this quantity still represents a significant computational challenge, which we leave for future work. "
Author: Albertus de Klerk on 2021-02-16 [id 1242]
(in reply to Report 2 on 2021-01-19)We thank the referee for their careful reading, and agree with the summary of the paper. Furthermore, we thank the referee for their support for our work.
In their report the referee mentions the following:
"It is not clear that the method can be extended to more than a dozen particles, which severely restricts its applicability."
Although we agree that applying the method to more than a dozen particles will be challenging for certain quenches, we would like to emphasize that the number of particles we are able to consider depends sensitively on the change in the interaction strength considered. The results presented in the paper for 6 particles comprise a worst case scenario, which was chosen due to the existence of the analytical results available for this case. For example, for the quench where the interaction strength is changed from 20 to 10, we can achieve great accuracy with modest computational effort, as shown in section 4.3.2.

---

## Round 2 · Referee Report · Anonymous (Referee 2) · 2021-1-19

Strengths
1. Novel method to solve 1d Bose gas for nontrivial interactions
2. Very detailed study of convergence
3. Method can be extended to a large claass of other systems
Weaknesses
1. For the Bose gas, the number of the particles that can be treated with the method is relatively small. It is not clear that the method can be extended to more than a dozen particles, which severely restricts its applicability.
Report
This paper introduces a new version of the truncated Hamiltonian method for the Bose gas. The main advance consists of a different truncation scheme, which prioritizes eigenstates with high overlap with the initial state, together with an improved renormalisation group method. The paper is a proof-of-principle investigation, and demonstrates a substantial improvement in convergence and accuracy. Despite the limitation in particle number, the results represent a significant and non-trivial advance over previously used approaches. I consider the paper fully suitably for Scipost Physics, and recommend its publication in the present form.
Requested changes
No changes requested.

---

## Round 3 · Referee Report · Anonymous · 2021-2-24

Report
I am really surprised by the authors reply that clearly did not read properly my report. I am still very much in favour of publication in SciPost physics, but what the authors wrote in their reply and added to the paper is a complete misunderstanding of my comment.
Calculating ab-initio the entanglement entropy, as the authors write, is so obviously a hopeless task that nobody will dare to do or to mention. I just wrote to use the knowledge of the root density and to plug in the quasiparticles picture formula, to be clear Eq. (4) of the paper https://arxiv.org/pdf/1712.07529.pdf (as one of the many places where this formula can be found). This not only is doable, but it is basically already done in the manuscript.
Requested changes
The authors should remove the comment about the ab-initio evaluation of the entanglement entropy and possibly add one about its quasiparticle determination.
Author: Albertus de Klerk on 2021-09-07 [id 1740]
(in reply to Report 2 on 2021-02-27)We apologize to the referee for misunderstanding their comment and thank the referee for their clarification of the proposed approach.
Approaching the problem of computing the time-evolution of the entanglement entropy using the existing approaches for the thermodynamic limit is indeed an interesting idea, and probably the only viable approach. However, we would like to note that the application of the formula referred to by the referee is strictly only valid after taking the thermodynamic as well as the scaling limit. Therefore, an investigation into the magnitude of finite size effects is warranted in order to understand to what extent the formula mentioned by the referee can be applied to our results. Due to the lengthy nature of our paper, we think that it would be better to leave this direction of investigation to future works.

---

## Round 3 · Author Response

List of changes
To address the report of referee 1 we have included the following comment in the conclusions section of the paper.
"Finally, we would like to point out that the method developed in this paper provides, in principle, all the ingredients necessary to compute for example the time evolution of the entanglement entropy. However, the computation of this quantity still represents a significant computational challenge, which we leave for future work. "

---

## Round 3 · List of Changes

To address the report of referee 1 we have included the following comment in the conclusions section of the paper.
"Finally, we would like to point out that the method developed in this paper provides, in principle, all the ingredients necessary to compute for example the time evolution of the entanglement entropy. However, the computation of this quantity still represents a significant computational challenge, which we leave for future work. "

---

## Round 4 · Author Response

With this resubmission we hope to clear up any misunderstanding there was regarding the computation of the entanglement entropy between us and the first referee.

---

## Round 4 · List of Changes

To address the response of referee number one to our changes we have modified our comment in the conclusions regarding the computation of the entanglement entropy to the following:

"Finally, we would like to point out that the method developed in this paper provides, in principle, all the ingredients necessary to compute for example the time evolution of the entanglement entropy. In order to come to a tractable computation one can convert the overlaps coming from the NRG-routines to a root distribution and then use the quasi-particle picture formulas for the entanglement entropy, see e.g. \cite{calabrese_evolution_2005,alba_entanglement_2018}. However, in order to ascertain the accuracy of results obtained in this way, a careful quantitative study of finite-size effects is required in order to determine if we can accurately match results in the thermodynamic and scaling limits. We leave addressing this challenge to future work."

---

## Editorial Decision

published